# Plutonium concentrations link soil organic matter decline to wind erosion in ploughed soils of South Africa

Joel Mohren[1,2]*, Hendrik Wiesel[3,4], Wulf Amelung[5], L. Keith Fifield[6], Alexandra Sandhage-Hofmann[5], Erik Strub[3], Steven A. Binnie[1], Stefan Heinze[7], Elmarie Kotze[8], Chris Du Preez[8], Stephen G. Tims[6], Tibor J. Dunai[1]

[1]Institute of Geology and Mineralogy, University of Cologne, Zülpicher Str. 49b, 50674 Cologne, Germany.
[2]Chair of Physical Geography and Geoecology, RWTH Aachen University, Wüllnerstr. 5b, 52062 Aachen, Germany
[3]Division of Nuclear Chemistry, University of Cologne, Zülpicher Str. 45, 50674 Cologne, Germany.
[4]Advanced Nuclear Fuels GmbH, Am Seitenkanal 1, 49811 Lingen, Germany.
[5]Institute of Crop Science and Resource Conservation, Soil Science and Soil Ecology, University of Bonn, Nussallee 13, 53115 Bonn, Germany.
[6]Department of Nuclear Physics and Accelerator Applications, Research School of Physics. The Australian National University, Canberra, ACT 2601, Australia.
[7]CologneAMS, Institute of Nuclear Physics, University of Cologne, Zülpicher Str. 77, 50937 Cologne, Germany.
[8]Department of Soil, Crop and Climate Sciences, University of the Free State, P.O. Box 339, Bloemfontein 9300, Republic of South Africa.

*Correspondence to*: Joel Mohren (joel.mohren@uni-koeln.de)

**Abstract.** Loss of soil organic matter (SOM) from arable land poses a serious threat to soil fertility and crop yields, and thwarts efforts to conserve soils as carbon sinks to mitigate global warming. Wind erosion can be a major factor in the redistribution of soil fines including SOM, but assessments of its impact have typically been limited by short observation periods of a few years at most. Longer timeframes, extending back to the mid 20th century, may however be probed using the concentrations of radionuclides that were globally distributed by nuclear weapon tests conducted during the 1950s and early 1960s. The basic concept is that differences in fallout radionuclide (FRN) activities between undisturbed and arable soils can be used to infer soil particle redistribution. In the present work, we have measured activities of $^{137}$Cs and $^{239+240}$Pu in soils from three agricultural regions of the plains of the South African Highveld. The three regions represent distinct agroecosystems and within each region the temporal length of cultivation varies from zero (i.e., native grassland) to almost 100 years. Wind erosion has previously been shown to play a dominant role in soil particle loss from agricultural sites in the Highveld, and the level plots we investigate here did not show any evidence of fluvial erosion. Hence, we interpret the fate of soil fines, including SOM, to be governed by wind erosion. For the cultivated soils, radionuclide activities are found to be less than in adjacent native grassland, and the magnitude of the reduction is strongly correlated with the duration of cultivation. Specifically, the original inventories of both $^{137}$Cs and $^{239+240}$Pu are approximately halved after ~20-40 years of cropping. The initial rate loss relative to the undisturbed soils is, however, considerably higher, with ~6% yr$^{-1}$ recorded during the first year after native grassland is converted to arable land. We correlate our radionuclide data with previously published SOM contents from the same sampled material and find that the radionuclides are an excellent indicator of SOM decline at the sites we investigate. We conclude that wind erosion can exert a dominant control on SOM loss in arable land of South Africa.

## 1 Introduction

### 1.1 Soil organic matter and its degradation

The net loss of soil organic matter (SOM) from arable soils poses a serious threat to the fate of human life on Earth. SOM content represents a core soil property, being a key determinant of soil fertility and therefore plant growth (e.g. Palm et al., 2007). Its decline leads to a reduction of crop production yields around the globe, threatening food security (e.g. Bot and Benites, 2005; FAO and ITPS, 2015). The SOM contents in arable soils can be reduced by a variety of factors, including biological, chemical, and physical soil degradation pathways (for an overview see Palm et al., 2007). The threats posed by unsustainable soil management practices, which may be amplified by anthropogenic climate change, have been highlighted in

many studies and summarized in major reports from international organisations (e.g. FAO and ITPS, 2015; FAO, 2017;
Reeves, 1997). In particular, aridity has been identified as the most significant factor in the degradation of agricultural land
(Prăvălie et al., 2021), and unsustainable cropping practices have been shown to affect soil fertility even under more favourable
(i.e., less arid) climatic conditions (Smith et al., 2016b). As a consequence, evidence is growing that wind erosion could play
an important role in SOM decline, especially in dryland regions, hence affecting both $CO_2$ budgets and crop yields (as e.g.
summarised in FAO and ITPS, 2015). The organic carbon storage in undisturbed soils tends to approach steady state conditions
over time, when inputs from net primary productivity and heterotrophic respiration outputs are in balance (Amundson et al.,
2015). Presently, however, unsustainable soil cultivation practices have unbalanced soil carbon fluxes on a global scale (e.g.
Sanderman et al., 2017). Wind erosion contributes significantly to loss of SOC from arable soil, both on regional (e.g. Chappell
et al., 2013; Yan et al., 2005) and global scales (Chappell et al., 2019). In particular arable systems on the African continent
are considered to be the most heavily affected by degradation on Earth (Prăvălie et al., 2021), and degradation by means of
wind erosion appears to play a key role in South Africa (Eckardt et al., 2020). The redistribution of soil particles by means of
wind can be quantified by dust collectors (e.g. Holmes et al., 2012), but there is a lack of empirical data resolving the fate of
SOM over longer timescales (Chappell et al., 2019).

## 1.2 SOM decline in the South African Highveld grasslands ecoregion

A hotspot of cultivation-induced decline of organic matter in arable soils are the arable plains located within the Highveld
grasslands ecoregion in the Free State province, Republic of South Africa. Here, various studies have investigated the impact
of soil cultivation practices on soil properties under semi-arid (e.g. Vos et al., 2020; Lobe et al., 2001) and temperate (e.g.
Lobe et al., 2001; Amelung et al., 2002) climate conditions. Within the Free State province, a huge contribution to land
degradation arises from deflation processes, which have been found to be largely a consequence of commercial cropping
introduced after the arrival of European settlers (Holmes et al., 2012). In particular surfaces in the semi-arid western portion
of the Highveld grasslands have been identified as dust sources, providing fines for more than 70% of all plumes recorded
between 2006 and 2016 (Eckardt et al., 2020). Similar scenarios have been identified in other countries around the globe (FAO
and ITPS 2015). The resulting reduction of soil fertility in the Highveld grasslands poses a threat to the food security in the
Republic of South Africa. The region contributes significantly to the crop production of the country; about one third of the
nation's field crop-cultivated land lies within the Free State administrative boundaries. Consequently, about 40% of maize
production, 41% of soybean production, and 20% of wheat production is concentrated in this region (numbers for 2021/22;
DALRRD, 2023). The majority of South African soils inherently have low SOM contents, emphasising the important role soil
protection plays in South African politics (a comprehensive review on the issue and related studies is given by Du Preez et al.,
2019). On the plot scale, knowledge on timing and magnitude of SOM content change due to erosional processes but also by
aeolian deposition (e.g. Dialynas et al., 2016) is of great importance to further propel efforts to minimise unsustainable
cropping practices.

In the beginning of the 21$^{st}$ century, Lobe et al. (2001) presented a dataset obtained from three agroecosystems located within
the Highveld grassland ecoregion to investigate the relationship between SOM content in arable sandy soils and the total time
period these soils were cultivated. The agroecosystems were named Harrismith (sample abbreviation HS, mean geographical
centre 28.4°S, 28.9°E), Kroonstad (KR; 27.9°S, 27.0°E), and Tweespruit (TW, 29.2°S, 27.1°E; Fig. 1, Table 1). The
agroecosystem concept provides a frame to group individual sites based on similar soil properties and environmental conditions
(cf. Du Toit et al., 1994). Besides grouping soil sampling sites, the sampling of Lobe et al. (2001) exclusively focused on level
(< 3°) plots, where no traces of fluvial erosion could be ascertained. Furthermore, accurate information on the time periods
over which the sites were subject to cropping was acquired, partially based on preceding studies of Du Toit et al. (1994) and
Du Preez and Du Toit (1995). From splits of the same sample material, consecutive studies analysed amino sugars (Amelung

et al., 2002), lignin compounds (Lobe et al., 2002), SOM $^{13}$C and $^{15}$N signatures (Lobe et al., 2005), soil aggregates (Lobe et al., 2011), as well as sulphur (Solomon et al., 2005) and phosphorous forms (Von Sperber et al., 2017). A key finding of the study published by Lobe et al. (2001) was that SOM contents decreased exponentially with increasing periods of cultivation. About 65% of SOM was lost after 90 years of cultivation (YOC), contemporaneous with a linear decrease of the silt content (2-20 μm) from about 10% to 5.4% (mean values; Lobe et al., 2001). Especially for SOC, the exponential decrease with increasing duration of cultivation was found to be most significant in the silt fraction, providing evidence for a significant contribution of wind erosion to SOM loss (Lobe et al., 2001). Comparable patterns of depletion were observed for amino sugars (Amelung et al., 2002) and oxidised lignin compounds (Lobe et al., 2002). From δ$^{13}$C analyses in SOM, Lobe et al. (2005) found the relative amount of grassland-derived SOM to be primarily reduced in the silt fraction, possibly as a consequence of selective removal in this fraction and a relatively higher input of organic matter from crops, when compared to other grain size fractions. Field evidence of deflation processes was e.g. reported by Lobe et al. (2011), who observed the accumulation of coarse soil particles close to topographic barriers at some rims of agricultural plots. About 100 km to the northwest of the Tweespruit sites, Wiggs and Holmes (2010) measured dust fluxes on a level (<2°) ploughed field belonging to the Grasslands farm near Bloemfontain. The authors reported a total dust deposition of 48.19 g cm$^{-2}$ (0.48 g m$^{-2}$ day$^{-1}$) from the local, wind-eroding sandy soils for at timespan ranging 99 days between August and November 2007. For the sites we investigate in our study, a re-assessment of the silt fraction content [partially unpublished, measured by Lobe et al. (2001) and Amelung et al. (2002); Table S1] reveals a linear increase in south-eastern direction (R$^2$ = 0.73; Fig. S1), which follows the general trajectories of dust plumes in South Africa (Eckardt et al., 2020).

## 1.3 Using fallout radionuclides to investigate the contribution of wind erosion to SOM decline in the South African Highveld grasslands ecoregion

As noted above, long-term quantitative information on the impact of wind erosion on the loss of SOM in southern Africa due to cropping is lacking to date. The means to overcome this problem is provided by fallout radionuclides (FRNs) from the atmospheric nuclear weapons testing in the 1950s and early 1960s. In particular, plutonium isotopes ($^{239}$Pu and $^{240}$Pu) and caesium-137 ($^{137}$Cs) were distributed world-wide. Concentrations of plutonium isotopes in soils can be measured with very high sensitivity using accelerator mass spectrometry (AMS; e.g. Fifield, 2008), and concentrations of $^{137}$Cs are determined by low-background γ-ray spectrometry (e.g. Wallbrink et al., 2003). In order to assess soil redistribution, FRN concentrations in undisturbed reference sites are compared with those in adjacent eroding sites (e.g. Zapata, 2002; Schimmack et al., 2002; Van Pelt, 2013). In the northern hemisphere, the use of $^{137}$Cs in this context has been seriously compromised by additional input from the Chernobyl accident in 1986 [see Meusburger et al. (2020) for a detailed study of $^{239+240}$Pu vs $^{137}$Cs inventories in Europe] and the Fukushima accident in 2011. The southern hemisphere was not affected, so $^{137}$Cs could still be used to complement the measurements of $^{239,240}$Pu in the present work. Compared to the plutonium isotopes, $^{137}$Cs has a rather short half-life of 30.08 yr [all decay values obtained from the U.S. National Nuclear Data Center (NNDC)]. At the time of measuring (2012), about two thirds of the $^{137}$Cs deposited during the atmospheric nuclear weapon tests conducted until the early 1960s had already decayed. Furthermore, the deposition in the southern hemisphere was less than one third of that in the northern hemisphere (UNSCEAR 2000). The concentrations of $^{137}$Cs were therefore approaching the detection limit of the γ-counting method, especially in heavily eroded soils and samples from depth. The plutonium isotopes, on the other hand, have much longer half-lives ($^{239}$Pu: 24,110 yr; $^{240}$Pu: 6561 yr), so losses due to decay are minimal. Consequently, plutonium is increasingly supplanting $^{137}$Cs as a tracer of soil redistribution (e.g. Alewell et al., 2017; Van Pelt and Ketterer, 2013).

The method to assess soil redistribution by using FRN concentrations relies on several assumptions which should be met (for an overview, see e.g. Van Pelt, 2013; Zapata, 2002; a critical assessment of the technique and a reply to the critical view are provided by Parsons and Foster, 2011, and Mabit et al., 2013, respectively). One precondition of the widely used traditional

sampling approach (cf. Li et al., 2011) is that of a homogeneous distribution of the target FRN over the limited area covering the undisturbed reference site and the nearby eroding sites. However, variability in wet and dry fallout deposition as well as microtopography even on the local scale needs to be considered. Thus, a rather short distance between the reference site and the eroding site and a large number of subsamples to characterise the reference site FRN inventory (i.e., $n > 10$; Sutherland, 1996) are considered as crucial (Sutherland, 1996; He and Walling, 1996; Van Pelt, 2013). A certain variance attached to reference inventories may be inevitable but can be reduced by applying the repeated-sampling-approach, which relies on on-site point-specific reference inventories (Li et al., 2011; Kachanoski and De Jong, 1984). Such a sampling strategy, however, would require a resampling campaign and hence be difficult to implement in our case given possible changes in land use and cropping practices since 1998 as well as individual permits required.

A reasonable application of the traditional approach relies on reference sites that ideally are vegetated with perennial grass or low herb cover (Pennock and Appleby, 2002) and shielded from sediment deposition, such as likely achieved on level upland sites (Funk et al., 2011). Once deposited on the soil surface, the migration behaviour of the fallout isotopes becomes important. In general, $^{137}$Cs and plutonium behave similarly in soils, as both are strongly adsorbed on soil fines, including SOM (e.g. Schimmack et al., 2001; Xu et al., 2013). However, evidence is growing that plutonium could have a greater sorption capacity to SOM than $^{137}$Cs (e.g. Schimmack et al., 2001; Alewell et al., 2017; Xu et al., 2017). While $^{137}$Cs sorption has been found to be generally dependent on the availability of cation exchange sites in soils and hence on clay mineralogy (Mabit et al., 2013; Parsons and Foster, 2011), it might bind more selectively to the clay fraction compared to plutonium, implying that $^{137}$Cs could be more sensitive to preferential transport (Xu et al., 2017). The potential migration pathway of plutonium as a solute is dependent on its oxidation state, with Pu(III) and Pu(IV) being considered the least mobile (Alewell et al., 2017; Meusburger et al., 2020). Apart from the soil type, rainfall regimes appear to affect the advection of plutonium isotopes, with sandy soils in arid environments showing potentially increased mobilisation as compared to clayey soils in the tropics (Cook et al., 2022). Vertical migration in the soil column can also be achieved due to physical processes, such as bioturbation or tillage. The latter, which is of particular importance to this study, has been shown to homogenise FRN concentrations throughout the A$_p$ horizon rapidly, e.g. after ~1-4 times of soil inversion (Schimmack et al., 1994; Hoshino et al., 2015). Similar as for $^{137}$Cs (Van Pelt, 2013; Mabit et al., 2013), plant uptake has been found to be insignificant for plutonium in natural settings (Harper and Tinnacher, 2008; Coughtrey et al., 1984), including grasslands (Little, 1980).

Given that the abovementioned conditions are met, soil redistribution rates may be derived from comparing the FRN inventories in eroding versus those found in reference sites. Most commonly, soil redistribution models, such as the linear proportional model or mass balance models, are applied (for a comprehensive overview, see Van Pelt, 2013). Usually, such models rely on high-resolution depth profiles from undisturbed soils (e.g. Meusburger et al., 2018; Lal et al., 2013; Alewell et al., 2014). The majority of such studies has focused on fluvial erosion, with less applications of $^{137}$Cs to quantify wind erosion and a substantial lack of wind erosion studies applying $^{239+240}$Pu (Alewell et al., 2017; Van Pelt, 2013; Van Pelt and Ketterer, 2013; Van Pelt et al., 2017). Most of the studies falling in the latter category identify wind as an erosional mode alongside fluvial erosion (e.g. Zhao et al., 2020; Liu and Hou, 2022), while very few address wind erosion as the main factor of soil redistribution (Little, 1980; Van Pelt and Ketterer, 2013; Michelotti et al., 2013; Van Pelt et al., 2017).

In this study, we use both $^{137}$Cs and $^{239+240}$Pu to reveal the proposed contribution of wind erosion to SOM loss in the three agroecosystems in the South African Highveld initially studied by Lobe et al. (2001) based on splits from original sample material taken in 1998. The samples encompass a wide range of cultivation histories, ranging from zero (i.e., native grassland) to 98 years. Our approach allows us to investigate the time evolution of SOM loss after native grassland is converted to cropland. Our study represents one of the first attempts to link plutonium activities to SOM loss by wind in arable lands. Furthermore, we are able to introduce a certain temporal resolution of process rates by analysing arable land with different cultivation histories.

## 2 Methods

### 2.1 Sampling strategy

The samples analysed in this study were taken in 1998 and splits from these samples have been measured in previous studies to investigate a variety of soil components and patterns of soil degradation over time (Sect. 1.2). In the following, we give a brief overview of the sampling strategy that was applied. Following the agroecosystem approach, the sites sampled within an agroecosystem were similar in soil type and environmental conditions (Table 1). Furthermore, sampling focused on upland agricultural areas with level surfaces to minimise the possibility of fluvial erosion and aeolian influx affecting the SOM content (and FRN concentrations). The sampling sites are located within the present-day local municipalities of Dihlabeng and Maluti a Phofung (HS), Matjhabeng (KR), and Mantsopa (TW). Each of the three datasets includes one composite sample (HS0, KR0, TW0) taken from native grassland sites located directly adjacent to the respective cultivated sites. These reference samples represent the amalgamated sample material from all grassland sites within a common agroecosystem. At these reference sites, soils had never been subject to ploughing and cropping by the time of sampling. The sites were grazed for up to three months per year by either cattle or sheep at stocking densities of ~0.5 large stock units per ha. At the cultivated sites, most farmers aimed for ploughing depths of 20 cm for cropping. Hence, the ploughing ($A_p$) horizons of the cultivated soils studied here were mostly 20-30 cm thick, except for two sites in the Harrismith agroecosystem, which were eventually ploughed to a depth of 40 cm (HS30 and HS68). By the time of sampling, none of the sites had ever been irrigated or organically fertilised. The soil pH values measured (~4.6-6.8; Lobe et al., 2001) rather imply that the least-mobile oxidation state Pu(IV) dominates plutonium inventories in the soils we investigate (cf. Alewell et al., 2017; Meusburger et al., 2020).

For each site, a radial sampling strategy following the suggestions of Wilding (1985) was applied. Five subsamples, taken by using a steel cylinder, were amalgamated per plot to obtain the final sample. Up to nine different agricultural plots were sampled per agroecosystem, with the requirement that the cultivation history (up to 98 yrs) could be precisely ascertained and a reference site could be sampled adjacent to the eroding site (cf. Lobe, 2003). Sampling focused on topsoils spanning the $A_p$ horizon (0-20 cm). A few samples were also collected at depths of 20-40 cm. The latter were included in order to test whether the topsoil sampling approach captured most of the plutonium stored within the soil column (cf. Parsons and Foster, 2011). The subsamples were taken at a horizontal distance of more than 3 m from each other. Differences in bulk densities between the sites were minor, not correlating with the duration of cropping, and explaining less than 1.3% of the differences in SOC contents between the sites (Lobe et al., 2001). Hence, no equivalent mass corrections were needed for accurate assessments of SOM and Pu loss rates.

The sampling scheme, which originally did not focus on FRN analyses, has some disadvantages potentially biasing FRN data interpretation (cf. Sect. 1.3). Firstly, the lack of high-resolution depth profile samples means that we are unable to present FRN mass depth profile data. Consequently, we cannot reasonably infer mass redistribution rates as typically presented in FRN studies (e.g. Alewell et al., 2014; Lal et al., 2013; Meusburger et al., 2018). Likewise, we cannot assess effects of soil particle fluxes that may alter inventories in the composite reference samples (cf. Chappell, 1999; Sect. 4.4), although visual inspection suggests that significant coarse-grained influx from local sources can be ruled out. Finally, amalgamation of the reference site samples ($n$ = 7-9 samples per agroecosystem; with $n$ = 5 subsamples per site) implies that we cannot provide statistical measures to evaluate the accuracy of fallout inventories in the reference samples.

### 2.2 Sample processing

The chemical sample preparation for plutonium followed the protocol of Everett (2009). The same protocol was employed for samples processed at the Department of Nuclear Physics, Australia National University (ANU), and the Division of Nuclear Chemistry, University of Cologne (UoC). The physical preparation of the samples was conducted at the Institute of Crop

Science and Resource Conservation, Bonn (sieving), and at the Commonwealth Scientific and Industrial Research Organisation (CSIRO), Land and Water Laboratories, Canberra (homogenisation).

In short, samples were sieved to obtain the <2 mm fraction and afterwards homogenised using a ring mill. For AMS, about 20 g per sample were dried at 105°C to constant weight. After adding a $^{242}$Pu spike (~5 pg, i.e. ~$10^{10}$ atoms, diluted from National Institute of Standards and Technology (NIST) Standard Reference Material® 4334H), the samples were dried overnight at 80°C and subsequently ashed at 450°C for 8 h. After ashing, the samples were leached for 48-72 h in 8 M $HNO_3$ at 90°C. Following the separation of leachate and soil, about 1 g $NaNO_2$ was added to the leachate to reduce the oxidation state from $Pu^{6+}$ to $Pu^{4+}$, which is retained by the BioRad AG®-1x8 resin used for the column chromatography (~3.5 g resin per column conditioned by 10 ml of 18 MΩ $H_2O$ and 20 ml 8 M $HNO_3$). After loading the sample solution on to the column, 30 ml of 8 M $HNO_3$ followed by 70 ml HCl (conc.) were added to elute contaminating elements. Plutonium was then eluted from the columns by adding 25 ml of warm (~40°C), freshly prepared HCl (conc.)/0.1 M $NH_4I$ solution. After taking the eluant to dryness, 2 ml $HNO_3$ (conc.) and 2 ml HCl (conc.) were added to the samples to remove iodine and $NH_4NO_3$. The solution was again taken to dryness at 110°C, and chloride removed by adding another 2 ml of $HNO_3$ (conc.). In order to provide sufficient bulk material for an AMS sample, 200 µl of a $Fe(NO_3)_3$ solution (~7 mg Fe $g^{-1}$) was added to the residue and subsequently taken to dryness at 120°C for 24 h or longer. The samples were then baked at 800°C for 8 h to convert the $Fe(NO_3)_3$ to $Fe_2O_3$, ensuring that the plutonium is uniformly distributed in an iron oxide matrix. In a final step, the samples were mixed with Ag (at ANU) or Nb (at UoC) powder at a ratio of 1:4 by weight and pressed into AMS targets. Replicates were prepared for most of the samples, always in separate batches. The masses of the samples and $^{242}$Pu spike are given in Table S2.

As a complement to the plutonium measurements, $^{137}$Cs was also measured for selected samples ($n$ = 12). From the Harrismith and Kroonstad agroecosystems, two topsoil samples were measured for each (HS0/0-20, HS45/0-20; KR0/0-20, KR40/0-20). The remaining measurements were conducted on topsoil samples (TW0, TW8, TW12, TW32, TW40, TW60, TW90) and one depth sample (TW60/20-40) taken from the Tweespruit agroecosystem. To measure $^{137}$Cs, 50-70 g of the same homogenised material used for AMS were pressed into cylindrical counting discs to ensure a well-defined geometry. These sample measurements were conducted at CSIRO. All $^{137}$Cs data presented in this publication have been decay-corrected to February 2012 (the time of measurement).

## 2.3 FRN measurements

All FRN measurements presented in this study were conducted in 2012. Initial plutonium measurements ($n$ = 16) were conducted at the ANU using the 14UD pelletron tandem accelerator of the Department of Nuclear Physics (Fifield, 2008). The accelerator was operated at ~4 MV, with $PuO^-$ ions injected. The molecular ions were dissociated and multiple electrons stripped from the plutonium atoms after the first stage of acceleration. This was accomplished by oxygen gas with a thickness of ~0.5 µg $cm^{-2}$ confined in an 80 cm long canal in the high-voltage terminal of the accelerator. After the second stage of acceleration, $Pu^{5+}$ ions with an energy of ~24 MeV were selected by the high-energy analysis system and detected in a gas ionisation detector with an energy resolution of 3%. At this resolution, plutonium ions are very effectively separated from interfering lower mass ions with the same mass per charge ratio. For example, $^{192}Os^{4+}$ ions, which would be accepted by the high-energy analysis system, have an energy that is 20% lower than $^{240}Pu^{5+}$ ions. The $^{239}$Pu/$^{242}$Pu and $^{240}$Pu/$^{242}$Pu ratios were measured, allowing the number of $^{239}$Pu and $^{240}$Pu atoms in the sample and the $^{240}$Pu/$^{239}$Pu ratio to be deduced. These ratios are also collected in Table 2, along with derived quantities from which the final concentrations are deduced. Quality control was ensured by regular measurements of United Kingdom Atomic Energy Authority (UKAEA) [62] Certified Reference Material (CRM) Pu 5/92138 with ratios $^{240}$Pu/$^{242}$Pu = 0.954(2%), $^{239}$Pu/$^{242}$Pu = 0.988(2%), and $^{240}$Pu/$^{239}$Pu = 0.966(1%). A similar procedure was employed for $n$ = 47 measurements at the CologneAMS (Dewald et al., 2013). The 6 MV Tandetron accelerator was operated at 2.9 MV, and ~13.6 MeV $Pu^{3+}$ ions were selected after acceleration. A secondary standard, prepared at the

Centro National des Aceleradores (CNA, Chamizo et al., 2015) and calibrated against the abovementioned CRM [62] by Christl et al. (2013), was employed. The isotope ratios for this secondary standard are $^{240}$Pu/$^{242}$Pu = 0.281(2.1%); $^{239}$Pu/$^{242}$Pu = 0.534(2.4%); $^{240}$Pu/$^{239}$Pu = 0.530(2.1%).

The $^{137}$Cs activities of the selected samples were measured by counting the characteristic 662 keV γ-rays. Well-shielded high purity germanium crystals (HPGe; Wallbrink et al., 2003) in the low level counting laboratory of the CSIRO Division of Land and Water were employed. Measuring times ranged between 1 and 9 days per sample.

## 2.4 Interpretation of $^{239+240}$Pu results

Propagated uncertainties provided for individual $^{239+240}$Pu concentrations are dominated by AMS counting statistics but include recorded weighing errors. From these $^{239+240}$Pu activities per mass (here also termed "specific activities") we derived inventories, i.e. activities per area, by including sampling depth and bulk density data (Table S1). The low scatter in soil densities of the pooled grassland samples (standard deviation of 0.04-0.12 g cm$^{-3}$) and the available grain size data indicate homogenous soil properties across the samples within the respective agroecosystems (Table S1). However, the slight variation in soil densities resulted in larger uncertainties of the pooled reference samples due to error propagation, when compared to the ploughed plots. An accurate sampling depth was achieved by using a 20-cm long steel cylinder for sampling, but we propagate a depth error of 0.5 cm for calculating the inventories.

In those cases where replicates were measured (predominantly at CologneAMS; HS0/20-40 and HS45/20-40 AMS targets were measured twice at the ANU, here considered as true sample replicates), the quoted plutonium concentrations are weighted means. The corresponding uncertainties are either dominated by internal errors (i.e., dominated by AMS counting statistics) as expressed by the weighted mean error, or by external sources of uncertainty. The latter are reflected by the standard error, i.e. the standard deviation of the set of measurements divided by the square root of the number of measurements $\sigma/\sqrt{n}$ (the larger uncertainty value was chosen for each sample). One outlier was identified in the CologneAMS replicate measurements of TW0/0-20 (TW0/0-20, batch COL-1; Table 2), which we excluded from further interpretation of the data. The measurement was about 70% below the specific activity measured for the other replicates of this sample (TW0/0-20 of batches COL-2 and COL-4, and batch ANU).

## 3 Results

### 3.1 Topsoil $^{239}$Pu and $^{240}$Pu activities

The blank included into the batch measured at the ANU implied a blank correction of <1% for the samples taken in the top 20 cm of soil. Only for the depth samples TW0/20-40 and TW60/20-40 was the blank subtraction significant, amounting to ~8% and ~4%, respectively. Deviations of CologneAMS and ANU AMS results ($n$ = 11 pairs measured in both facilities) are statistically insignificant at a significance level of 0.05 (p-value of 0.95; paired two-tailed t-test). Likewise, replicate measurements at CologneAMS reveal a good reproducibility of the obtained results, with an average error of about 5% with respect to the weighted means calculated for the CologneAMS measurements (one replicate outlier from TW0/0-20 excluded, Sect. 2.4).

The measured inventories in the top 20 cm of soil span a wide range between 0.43 ± 0.01 mBq cm$^2$ (KR98/0-20) and 1.95 ± 0.06 mBq cm$^2$ (TW0/0-20; Table 2; concentrations are presented in Table S3). Sample KR98/0-20 is from the plot with the longest cropping history, and the opposite is valid for the pooled native grassland sample TW0/0-20. Similarly, the other samples from the uncultivated plots in the other two agroecosystems also have the largest inventories in their respective agroecosystems (HS0/0-20 1.44 ± 0.08 mBq cm$^2$; KR0/0-20 0.98 ± 0.03 mBq cm$^2$).

We calculate the measured $^{239+240}$Pu inventories for all three agroecosystems to reflect relative inventory concentrations against the relevant pooled samples from the uncultivated plots. As a function of the duration of cultivation, a trend of initially decreasing activity with increasing cropping time is evident, although the rate of decline slows as time goes on (Fig. 2). The data may be fitted ($R^2 = 0.76$) to a single exponential plus a constant expression in the form

$$I(t) = I_{eq} + (I_0 - I_{eq})e^{-t/\tau}, \tag{1}$$

where $I_0$ is the initial concentration (i.e. the uncultivated value; 100% by definition), $I_{eq}$ denotes an equilibrium value, and $\tau$ is the time constant for the decline of plutonium with time of cultivation (cf. Lobe et al., 2011). Sample KR2.5/0-20 shows an elevated relative inventory of 103.84 ± 4.22% (relative concentration 99.93 ± 3.00%) but does overlap within uncertainties with the defined initial activity. Hence, the sample was excluded from the fit. From the fit, $I_{eq}$ equals 56.03 ± 6.01% (1$\sigma_x$), and $\tau$ equals 6.86 ± 3.03 years.

### 3.2 Plutonium source and cross-validation with $^{137}$Cs

The $^{240}$Pu/$^{239}$Pu ratios calculated from the atomic ratios of $^{240}$Pu/$^{242}$Pu and $^{239}$Pu/$^{242}$Pu measured both at CologneAMS and ANU are consistent (Fig. S2). The weighted mean of the measurements for the 0-20 cm samples is 0.180 ± 0.002. Measured $^{137}$Cs activities (similar to $^{239+240}$Pu, all $^{137}$Cs measurements were performed in 2012) range between 2.04 ± 0.17 mBq g$^{-1}$ (TW0/0-20) and 0.40 ± 0.08 mBq g$^{-1}$ (KR40/0-20). The $^{137}$Cs ($A_{Cs}$) and $^{239+240}$Pu ($A_{Pu}$) specific activities correlate linearly ($R^2 = 0.97$; Fig. 3), with

$$A_{Pu} = 0.03A_{Cs} + 0.05. \tag{2}$$

Sample HS0/0-20 has been excluded for the derivation of Eq. (2), as it was identified as an outlier (Sect. 4.1; Table 2). Furthermore, the depth sample TW60/20-40 did not yield meaningful $^{137}$Cs data, as the specific activity of $^{137}$Cs was below the detection limit of the γ spectrometer. The weighted mean $^{137}$Cs/$^{239+240}$Pu ratio is 26.69 ± 0.97 (Table S4).

### 3.3 Plutonium inventories in depth samples

Ploughing depths in the studied region were usually 20 cm, resulting in the formation of a ploughing horizon A$_p$ of 20-30 cm thickness. In order to investigate whether plutonium could have migrated below this soil layer (e.g. Parsons and Foster, 2011), samples spanning the depth interval 20-40 cm were analysed for selected sites from the Tweespruit ($n = 4$) and Harrismith ($n = 2$) agroecosystems. The results indicate that inventories are generally much lower than in the top 20 cm, ranging from ~5 to 36% of what is measured in the corresponding topsoil sample (Table 2; Fig. 4). Sample HS45/20-40 is, however, a conspicuous exception with a surprisingly high inventory of 1.01 ± 0.03 mBq cm$^2$ in the 20-40 cm interval, which is even higher than the 0.70 ± 0.03 mBq cm$^2$ measured in the uppermost 20 cm of the soil (HS45/0-20).

## 4 Discussion

### 4.1 FRN concentrations in soils of the Highveld ecoregion

The $^{240}$Pu/$^{239}$Pu ratios indicate that plutonium activities measured in soils from the South African Highveld originate predominantly from global fallout. The average ratio of 0.180 ± 0.002 falls comfortably within the range of 0.173 ± 0.014 determined by Kelley et al. (1999) for the southern equatorial (0-30°S) region. Further support for a global fallout origin is provided by the $^{137}$Cs data. The weighted mean $^{137}$Cs/$^{239+240}$Pu ratio of 26.69 ± 0.97 (measured at ~27-30°S) is similar to that measured (24.2 ± 1.3) by Everett et al. (2008) at 18.4°S in the Herbert River catchment in Queensland, Australia. In addition, Bouisset et al. (2018) used data provided by other studies (Hardy et al., 1973; UNSCEAR 2000) as well as their own to

calculate a mean ratio ~25 for the southern hemisphere (all cited data decay-corrected to 2012). In line with this argumentation, the $^{239+240}$Pu inventories obtained from the native grassland composite samples are in the range expected for surface samples located within 20-30°S, which has been constrained to be $1.44 \pm 0.59$ mBq cm$^{-2}$ (Hardy et al., 1973).

In contrast to $^{137}$Cs, Pu isotopes can be measured on AMS systems with extremely low background (Fifield et al., 1996; Everett et al., 2008; Fifield, 2008). Our plutonium data indicate a high degree of measurement precision, with an average uncertainty attached to the individual samples measured in replicate being below 5%. However, the $^{137}$Cs measurements can help to further evaluate the $^{239+240}$Pu data (Fig. 3). Equation (2) predicts a minor excess of $^{239+240}$Pu activities ($5.4 \pm 1.9$ mBq kg$^{-1}$) as compared to $^{137}$Cs activity in the soils. Exceeding $^{239+240}$Pu has been proposed to reflect grain-size dependent preferential adsorption

patterns (e.g. Everett et al., 2008, Xu et al., 2017), and such a pattern could become important in case of selective erosion or soil particle influx. For our datasets, however, the offset we observe appears to be insignificant given the wide range of other factors that could have influenced the measurements (e.g., background subtraction from the γ-ray measurements). Sample HS0/0-20, i.e. the composite sample of native grasslands in the Harrismith agroecosystem, is the only conspicuous exception to the otherwise linear correlation between $^{137}$Cs and $^{239+240}$Pu concentrations. We cannot offer an explanation for the

observation, but note that such deviations are not uncommon in earlier work, with both unexpectedly high and low $^{137}$Cs activities (e.g. Xu et al., 2013; Bouisset et al., 2018; Fulajtar, 2003). In general, the linear correlation between $^{137}$Cs and $^{239+240}$Pu activities supports the assumption that chemical erosion does not contribute to the loss of the FRNs in the soils we investigate (cf. Everett et al., 2008). However, any leakage could equally affect the isotope concentrations and may thus not be reflected by the ratio.


## 4.2 The fate of $^{239+240}$Pu in topsoils after native grassland is converted to arable land

In most undisturbed soils, the majority of the plutonium originating from global fallout is stored in the uppermost centimetres of the soil column (e.g. Alewell et al., 2017; Hoo et al., 2011; Lal et al., 2020). The small amount of plutonium in the 20-40 cm depth interval for the uncultivated HS0 and TW0 samples would indicate the same is true within the agroecosystems we

investigate. The plutonium concentrations we measure in our samples from native grassland represent upper limits for $^{239+240}$Pu within the respective agroecosystems. Hence, these samples can be used as reference to assess the decline of plutonium concentrations in adjacent, cultivated plots over time relative to the reference sites.

In the cultivated soils, however, ploughing will distribute the plutonium-marked soil particles across the ~20 cm ploughing horizon (e.g. Fulajtar, 2003). With the exception of the anomalous HS45/20-40 sample (see sect. 3.3), the plutonium

concentrations measured from the depth samples generally support the conclusion that the majority of $^{239+240}$Pu was indeed stored in the upper 20 cm of the soil columns by the time of sampling (Fig. 4). It is noteworthy that HS45 is one of two samples where the bulk density of the depth sample was lower than the bulk density of the corresponding topsoil sample (1.24 g cm$^{-3}$ vs. 1.31 g cm$^{-3}$; Table S1). In addition, Harrismith is the only agroecosystem where also ploughing to 40 cm depth had been reported, although not recorded by us for this specific sample. Bioturbation can result in a further relocation of topsoil particles

to depths exceeding the A$_p$ horizon, but overall activities would be expected to decrease rather than increase with depth (Fulajtar, 2003). Since our sampling strategy included a spatial averaging of sampling material from each plot investigated (Sect. 2.2), the best explanation for the elevated plutonium activity in HS45/20-40 may be related to a former ploughing to 40 cm near in time to peak plutonium deposition that was not recorded during farmers' interviews or to sample contamination. Elevated inventories measured in two further depth samples might point to a certain degree of leakage of plutonium towards

greater depths, but not necessarily in the pre-fallout soils (Sect. 4.3). We also note that in a reasonably comparable setting (Bsh climate; Big Spring, USA), $^{137}$Cs concentrations dropped sharply below the A$_p$ horizon in an Aridic Paleustalf that had been cultivated since 1915 (Van Pelt et al., 2007; Van Pelt and Ketterer, 2013).

Our own plutonium data indicate that the greatest loss of soil particles occurs during the first years after the conversion from native grassland to arable land. Equation (1) predicts a decline in the $^{239+240}$Pu inventory of ~6% to ~2% per year during the first 10 YOC. After ~20-40 years, the measured inventories approach the equilibrium level at ~56% of the initial reference values, here constrained by a drop in the decline rate below 0.1% per year. Hence, Eq. (1) predicts that about half of the initial aerial activities in the soil is retained over the long term. The finding of an exponential decline over time towards the equilibrium asymptote indicates that a certain fraction of plutonium adsorbs to soil particles that appear to be shielded against wind erosion. The reason for this pattern could be linked to soil aggregation. For the sites we investigated, Lobe et al. (2011) found an equilibrium soil content of more than 60% for aggregates >250 μm after ~17-31 YOC. Such particle sizes are not likely to be suspended by wind action, achieving a certain level of protection for aggregated fines. The change in the distribution of soil aggregate fractions within the soils was generally found to have approached an equilibrium after ~31 YOC, mimicking the fate of $^{239+240}$Pu activities (Lobe et al., 2011). However, aggregation was shown to be least developed in the Kroonstad agroecosystem due to a greater abundance of coarse-grained particles (soil types defined as loamy sand, Lobe et al., 2011; see also Table S1). In this agroecosystem, our related plutonium inventories do often significantly exceed the trend predicted by the exponential model (Fig. 2A), indicating that other mechanisms could be at play affecting the primarily wind-governed decrease of $^{239+240}$Pu in arable land over time, some of which we discuss below.

### 4.3 Factors that may influence the interpretation of $^{239+240}$Pu topsoil inventories

The plutonium isotopes measured in this study allow us to reconstruct the fate of soils after native grassland is converted to arable land in the South African Highveld. However, the approach has a temporal limitation inherent to the application of anthropogenic FRNs for tracing the redistribution of soil particles. Since fallout occurred as consequence of atmospheric nuclear weapon tests, the timescales we can resolve is on its lower end limited by the major episode these tests were conducted, i.e. the mid 1950s to early 1960s (UNSCEAR 2000). Arable land that has been subject to ploughing before that time (we use the year 1963 as anchor point, i.e. the year the fallout of FRNs peaked on the southern hemisphere, UNSCEAR 2000) is very likely to have faced erosion before the FRNs were deposited. The relative $^{239+240}$Pu inventories obtained from arable land with a cultivation history exceeding 35 years mostly plot below the equilibrium asymptote defined by Eq. (1) in Fig. 2A, i.e. the weighted mean of these data points is 39.2 ± 3.7% (arithmetic mean 46.2 ± 11.0%). The low scatter of these points is noteworthy because apart from the possibility of varying environmental conditions prevailing even across short spatial distances (e.g. Smith et al., 2016a; Sutherland, 1996; He and Walling, 1996), other factors appear to be realistic that potentially could have significantly affected the deposition of fallout nuclides on the sites we investigated. For example, plutonium was deposited during the 1950s and 1960s on soils that by that time had been already subject to cultivation practices for decades. Consequently, soil properties at these sites had been already altered, with SOM contents being decreased (Fig. 2B) and/or soil texture significantly changed. Another possible factor that may lower plutonium concentrations in the >35 YOC plots is linked to fallout interception by plants. Adsorption of FRNs on plant surfaces, including crops, has been found to be potentially significant (e.g. Nakanishi, 2013; Reissig, 1965), and thus a certain proportion of the introduced plutonium could have been lost during crop harvest during the early 1960s. Likewise, a possible incorporation of Pu-marked plant material into the soil column after harvesting might have contributed to elevated inventories found in the three depth samples with cropping histories exceeding 35 years (HS45/20-40, TW40/20-40, and TW60/20-40), possibly indicating a certain leakage of plutonium-marked soil particles below the topsoil in these samples. In case of wet fallout deposition, such an enhanced downward migration could also have been promoted by the physical disturbance of the ploughed soil (cf. Das Gupta et al., 2006). These factors appear not to have been significant for the plot that was converted after the peak episode of global fallout (TW32/20-40). However, the topsoil samples TW40/0-20 and TW60/0-20 had the strongest inventory losses of all samples (Fig. 2a; Table 2), and show the highest negative residues against the exponential model. Thus, and given the general low scatter of the post-35 YOC data

points, we may argue that if significant migration of Pu-marked soil particles below 0-20 cm has occurred, the two samples in question could represent the cases of maximum leakage in our dataset.

In the topsoil, occasional grazing may also have contributed to a loss of FRNs, but given the low stocking densities more likely due to the ingestion of fallout-exposed grass (cf. Sato et al., 2017) than by means of wind erosion. These mechanisms may explain the minor trend of dropping $^{239+240}$Pu concentrations in some plots exceeding 35 YOC. However, in general terms the soil retention capabilities with respect to $^{239+240}$Pu adsorption appear to have been similar in native grasslands and more degraded cultivated soils by the time of FRN deposition. Another possibility factor affecting the overall decline of plutonium concentration over the long term could be related to influx of Pu-marked soil fines to the investigated sites (Sect. 1.2 and 4.4), although we exclusively focused on upland areas.

An approach to overcome the problem of a point source in time determined by anthropogenic global fallout involves the quantification of excess $^{210}$Pb, hereafter referred to as $^{210}$Pb$_{ex}$. This naturally occurring FRN has been widely used for erosion studies elsewhere (e.g. Matisoff, 2014; Mabit et al., 2008; Hu and Zhang, 2019; Meusburger et al., 2018), and was measured at CSIRO alongside $^{137}$Cs (Table S6). From this data, general trends of concurrently decreasing $^{210}$Pb$_{ex}$ activities with decreasing $^{137}$Cs activities (Fig. S3) and SOC content (Fig. S4) might be inferred. However, the short half-life (~22.2 yrs) of the nuclide plus >10 years of sample storage and generally low environmental concentrations cause propagated measurement errors of 16-30% of the respective means. Hence, we refrain from providing a detailed interpretation of $^{210}$Pb$_{ex}$ activities in our samples.

## 4.4 Adsorption of $^{239+240}$Pu to SOM and deflation of SOM

Despite the temporal limitation inherent to the application of anthropogenic FRNs, it is evident that $^{239+240}$Pu concentrations and SOM contents behave similar after native grassland has been converted to arable land. Bulk SOC has been shown to approach an equilibrium concentration of $39.4 \pm 2.0\%$ of the initial values after $33.6 \pm 8.0$ YOC, with $\tau = 6.7 \pm 1.7$ years (Fig. 2B; Lobe et al., 2001; their exponential model). The SOC equilibrium concentration differs from the Pu inventory equilibrium (and, likewise, the Pu concentration equilibrium, Fig. S5). This deviation may to a certain extent be explained by the cut-off of Pu data included in the fit after 35 YOC (Sect. 4.3). On the other hand, the similarity in time constants might indicate that the plutonium was similarly distributed as SOC in the upper 20 cm of the soil soon after initial ploughing, as homogenisation appears to be generally achieved rapidly (Sect. 2.1). Consequently, SOM contents and Pu activities both reflect largest rates of decline during the first years after native grasslands were converted to arable land. This similar behaviour of $^{239+240}$Pu activities and SOM content over time indicates a strong linkage between both variables. The relationship is underscored by high correlation coefficients ($R^2 = 0.56$ to $0.99$; Fig. 2C) between the plutonium concentrations (1963-1998) and bulk soil SOC contents as measured by Lobe et al. (2001) (Table S5). Similar $R^2$ values ($0.63 – 0.99$) are obtained when $^{239+240}$Pu is correlated to total N contents (Fig. S6).

Lobe et al. (2001) identified the turnover of SOM in the silt and clay fractions as the main factor controlling the time-dependent decrease of SOM stocks in the investigated arable soils. Mineralisation processes were believed to play an important role for the overall SOM decline, as expressed by the approach of lower SOM equilibrium levels (e.g. Lobe et al., 2001; Du Preez and Du Toit, 1995; Du Toit et al., 1994; Lobe et al., 2002). The approach of a time-independent SOM equilibrium has been attributed to the mineralisation-driven depletion of the labile SOM pool, thereby increasing relative dominance of the stable SOM pool in the soils (Lobe et al., 2001). Another contribution to the SOM decline was assumed to be caused by a strongly reduced input of organic matter during the fallow (Amelung et al., 2002; Lobe et al., 2001; Lobe et al., 2002), as well as due to the loss of silts via wind erosion, which likely also contained old oxidised lignin (Lobe et al., 2001; Lobe et al., 2002). Here, we indeed observe a strong correlation between plutonium activities and SOM contents, and the majority of SOM (80-90%;Lobe et al., 2001) is bound in the <20 μm fraction of the soils. Likewise, Plutonium isotopes are generally considered to

be predominantly bound in that soil fraction (Xu et al., 2017). Hence, we find it reasonable to assume that $^{239+240}$Pu is not likely to reflect a dominance of mineralisation processes but that SOM contents are primarily also affected by deflation processes rather than turnover rates. This finding is in line with similar studies who highlighted the significance of SOC removal due to wind erosion on different spatial scales (e.g. Chappell et al., 2013; Chappell et al., 2019; Yan et al., 2005).

It should be noted that the close correlation between bulk SOM loss and Pu activity decrease includes SOM losses due to particulate organic matter (POM). Due to its lower density, a certain degree of POM loss by wind erosion is more likely than for similar-sized clastics. Besides, the fate of mineral associated organic matter and POM can be closely coupled (Lobe et al., 2001; Lavallee et al., 2020; Sokol et al., 2019), and the formation of new SOM from the crops was found to be limited at the sites we investigate (Lobe et al., 2005). Consequently, wind erosion may have altered internal steady-state equilibria that control the portioning of SOC between POM and mineral surfaces (cf. Cotrufo and Lavallee, 2022; Lützow et al., 2006). With the latter being lost by erosion, possibly also the pressure on POM degradation increased, particularly if POM was to some extent protected by minerals.

This hypothesis could thus be to a certain extent modified by considering soil aggregation (Sect. 4.2). Breaking down soil aggregates as a consequence of ploughing could make SOM available for decay which would otherwise be more resistant to chemical oxidation (Elliott, 1986). While we do not assume that $^{239+240}$Pu concentrations will reflect this mineralisation mechanism directly, the supply of soil fines to be carried away by wind erosion could mimic the mineralisation loss rate of previously protected SOM. Furthermore, a certain degree of leakage of plutonium-marked particles to greater depths could contribute to a lowering of $^{239+240}$Pu activities in the topsoil over time. These mechanisms, together with the linear decrease of silt contents (no trend observable for the clay fraction) in cropped plots as observed by Lobe et al. (2001) (Figs. S1, S7), may imply that $^{239+240}$Pu activities will not approach an equilibrium in plots older than 35 years as predicted by Eq. (1). Instead, a bi-exponential model as suggested to predict SOM decrease over time (e.g. Amelung et al., 2002; Lobe et al., 2001; Lobe et al., 2011) could reflect the long-term fate of $^{239+240}$Pu inventories in the topsoils more accurately. To resolve this issue, longer timescales need to be monitored and high-resolution depth profiles sampling implemented in future studies.

A further mechanism that could be influencing SOM contents and absolute Pu activities may be related to deposition processes (cf. Chappell et al., 2014). Eckardt et al. (2020) concluded that the vast majority of dust plumes in South Africa have trajectories towards the southeast, approaching the Indian Ocean. In our study, we measure the lowest activities in the upwind sites. The composite grassland sample HS0/0-20 has an activity twice as high as the most north-west located sites (KR0/0-20), generally coinciding with SOM patterns as published by Lobe et al. (2001). For FRNs, such differences could arise from spatially variable deposition patterns, given the distances between the different agroecosystems of about 100-300 km. However, grain size data also indicate an increase in the silt fraction towards the south-east (Fig. S1). Consequently, KR soils had the lowest silt fraction of all soils (Fig. S7) by the time of sampling. These differences between the three agroecosystems may not only be attributed to slightly differing soil properties (Table 1, S1) but to regional environmental conditions and atmospheric circulation patterns, as found elsewhere (e.g. Xu et al., 2017; Meusburger et al., 2020; Funk et al., 2011). If such regional patterns of sediment redistribution caused significant influx of soil particles to reference sites from both local and regional sources after global fallout (cf. Wiggs and Holmes, 2010), it is possible that FRN inventories have been subject to alterations. As our methodological approach relies on undisturbed reference sites, significant influx to the reference sites would violate that most important precondition. Chappell (1999) showed that influx of soil particles can significantly alter $^{137}$Cs specific activities at a reference plateau site located in semi-arid bushland. Influx of distal dust particles may increase FRN concentrations (Chappell, 1999), more proximal influx of coarser grains could dilute them (Funk et al., 2011). Hence, post-fallout accumulation of soil particles on our reference sites could have different effects on the overall FRN concentrations, depending on the concentration of FRNs in the deposited soil particles (generally linked to soil particle grain size and source). However, visual inspection of the reference sites before sampling suggested that significant coarse-grained influx from local sources can be ruled out. In addition, Funk et al. (2011) demonstrated that $^{137}$Cs reference sites rather unaffected by aeolian

deposition could be identified in their study region, which resembles our study setting (grassland plateau site in Mongolia with significant wind erosion). We also note that soil bulk densities are generally homogenous across the individual agroecosystems and that the $^{137}$Cs and $^{239+240}$Pu concentrations we obtained are strongly correlated (Fig. 3). Given that $^{137}$Cs and $^{239+240}$Pu are suspected to show somewhat different grain size and SOM-dependent adsorption patterns (Sect. 1.3), the finding could imply that influx to the reference sites was limited. This hypothesis is supported by the fact that Furthermore, given that reference

sites were located directly adjacent to the eroding sites, alterations of the relative inventories due to soil particle influx should decrease in significance at decreasing YOCs.

**5 Conclusion**

We have measured fallout radionuclides ($^{137}$Cs, $^{239+240}$Pu) to quantitatively investigate the linkages between SOM decline and wind erosion in plain arable land of South Africa's Highveld grassland ecoregion. Wind erosion, a physical process, appears to be a dominant factor removing SOM particles from the plots we investigated. Here, wheat and maize yields have been reported to be more than halved after about 30 years of cultivation (Lobe et al., 2005). The severity of wind erosion can be promoted by the cropping practices that are commonly observed in the Free State province (Eckardt et al., 2020; Wiggs and

Holmes, 2010) and that were applied at the sites we investigated at least until the time of sampling. These cropping practices include the clearance of arable land from any vegetation for up to 6 months per crop rotation cycle (1-2 years) to minimise soil water loss by plant uptake during the dry season. Consequently, dust emissions peak during the winter months, when arable soils remain largely unprotected in South Africa's rainfed agriculture (Eckardt et al., 2020). Under the impression of anthropogenic climate change, which is in turn predicted to increase both drought and storm event probabilities southern Africa

(Arias et al., 2021), the data we present provides further evidence that these cropping practices cannot be termed sustainable. We find similar patterns of relative SOM decline in our investigated sites which are located in different agroecosystems at distances of hundreds of kilometres between each other. Based on the observations of Eckardt et al. (2020) and the grain size data obtained by Lobe et al. (2001) and Amelung et al. (2002), it appears reasonable to assume that SOM particles are conveyed to the Pacific Ocean. It is still a matter of debate whether oceans can be generally considered as sinks for organic carbon (for

an overview see Chappell et al., 2019), but it seems likely that SOC particles' exposition to decay is enhanced during transport in the atmosphere (e.g. Lal, 2006), contributing to negatively balance the worldwide carbon budgets. However, the effects may be less severe than they would have been if mineralisation controlled the release of $CO_2$, given the chance of carbon fixation in the ocean.


**Author contribution**

JM – Formal analysis, validation, visualisation, writing – original draft, writing – review and editing; HW – Formal analysis, investigation, validation, visualisation, writing – original draft; WA – Conceptualisation, funding acquisition, project administration, resources, supervision, writing – review and editing; LKF - Formal analysis, investigation, methodology,

project administration, resources, supervision, validation, writing – review and editing; ASH – Project administration, resources, supervision, writing – review and editing; ES – Conceptualisation, methodology, project administration, resources, supervision, writing – review and editing; SAB – Project administration, resources, supervision; SH – Investigation, methodology, resources; EK – Investigation, resources; CDP – Investigation, resources, writing – review and editing; SGT – Investigation, methodology, resources, supervision; TJD – Conceptualisation, funding acquisition, project administration,

resources, supervision, writing – review and editing.

**Competing interests**

The authors declare that they have no conflict of interest.

**Acknowledgements**

This work was funded by the Deutsche Forschungsgemeinschaft (German Research Foundation, DFG) in the frame of TRR228 (project A01). Claus Feuerstein is thanked for his successful efforts to improve the plutonium measurement capabilities at CologneAMS. Wulf Amelung additionally acknowledges support from the DFG under Germany's Excellence Strategy, EXC-2070-390732324-PhenoRob. The Heavy Ion Accelerator Facility (HIAF) at ANU is supported by the National Collaborative
Research Infrastructure Strategy (NCRIS) of the Australian Government. We thank three anonymous referees, whose comments have significantly improved the quality of this paper.

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

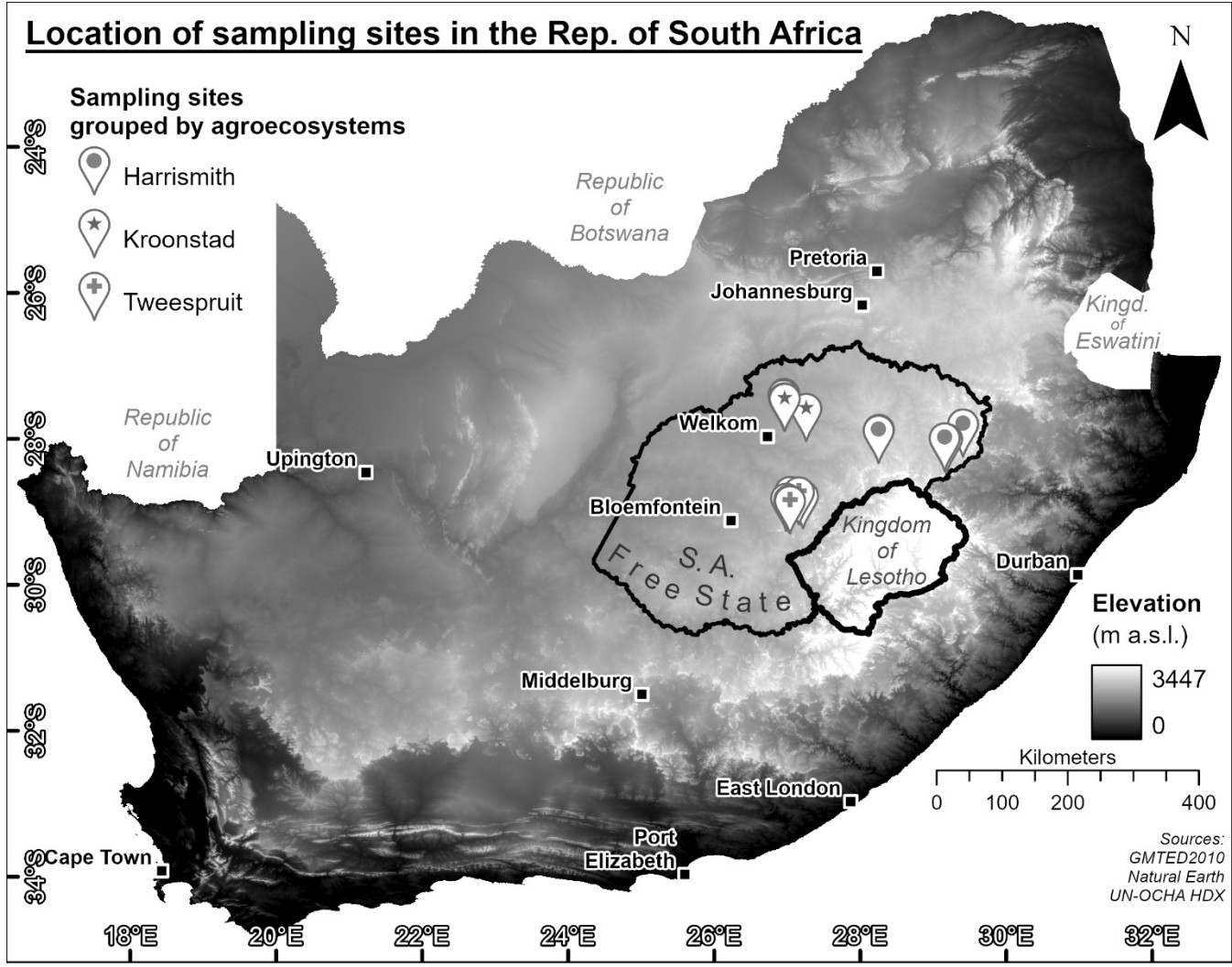

**Figure 1: Topographic map of the Republic of South Africa and the state territory of the Kingdom of Lesotho. Sampling locations are highlighted by tear pins. Administratively, the investigated sites are located within the Free State Province of the Republic of South Africa. The province largely stretches the Highveld ecoregion, which resembles an elevated, open grassland plain landscape.**

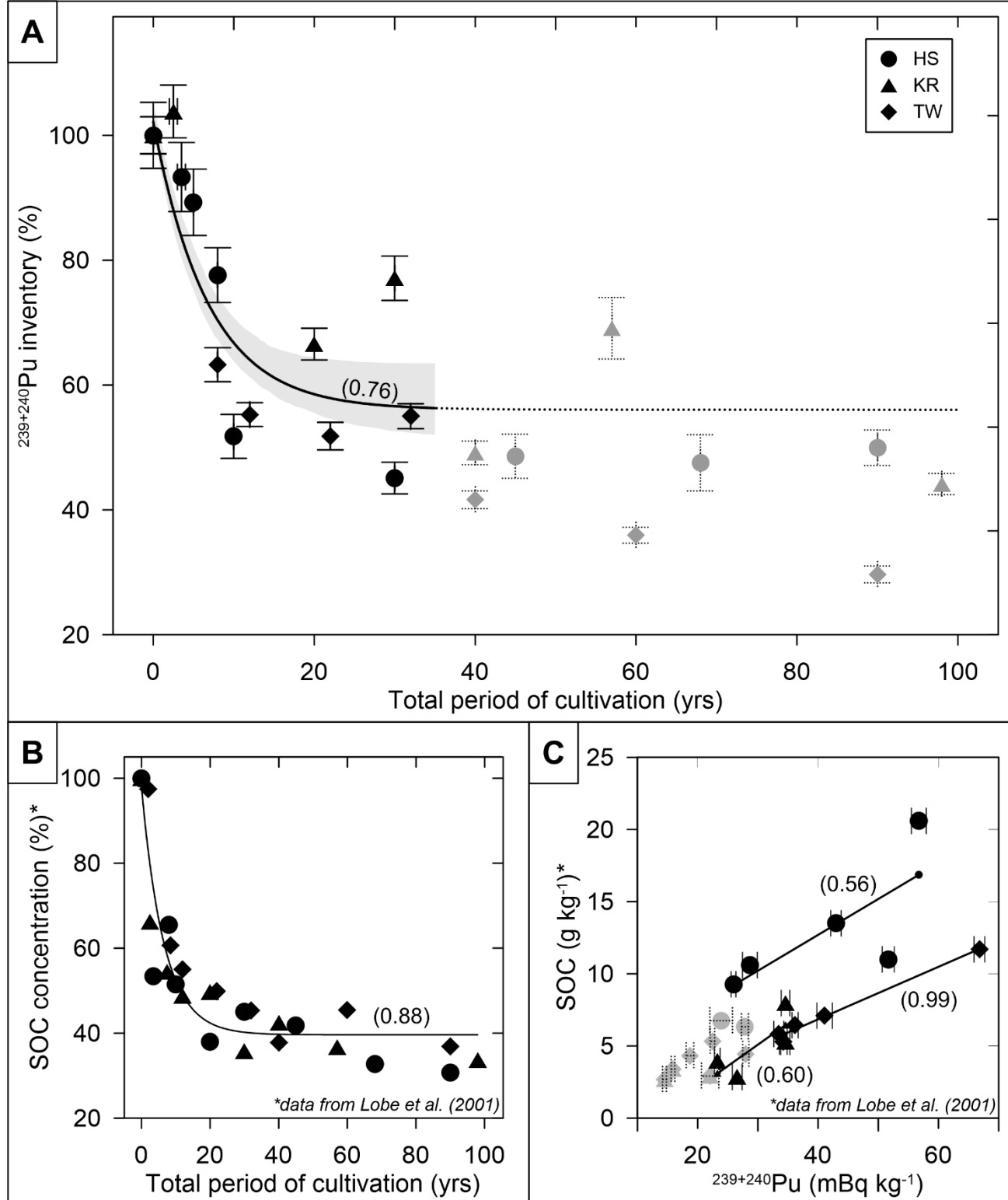

**Figure 2: Changes in topsoil fallout inventories (A) and bulk soil organic carbon (SOC) content (B; Lobe et al., 2001) over time, and correlation of the SOC and $^{239+240}$Pu concentrations (C). The inventories in cultivated soils are shown relative to those found in adjacent native grassland soils (i.e., 100% at t = 0). The mono-exponential regression (thin black line, enveloped by grey 68% confidence interval) in panel (A) indicates the approach towards a concentration equilibrium level after about 20-40 years of cropping. By that time, about 50-60% of the initial $^{239+240}$Pu inventory has been lost. The extrapolated post-35 years cropping equilibrium level is indicated by the dashed line. The relationship between $^{239+240}$Pu and SOC indicates that the decrease of SOC can be traced by measuring $^{239+240}$Pu in bulk soil (C). Most plutonium samples depict replicate measurements; the corresponding activities are weighted means and the uncertainties either dominated by AMS counting statistics (weighted mean error) or external sources of uncertainty (standard error). For single measurements, the 1σ measurement uncertainty provided by the AMS facilities dominates the final uncertainty. All greyed out data points with dashed error bars denote those samples that were taken from plots with more than 32 years of cultivation history (for discussion see text). Filled circles denote samples from the Harrismith (HS) agroecosystem; filled triangle those from the Kroonstad (KR) agroecosystem and filled diamonds those from the Tweespruit (TW) agroecosystem. Numbers in brackets denote correlation coefficients ($R^2$).**

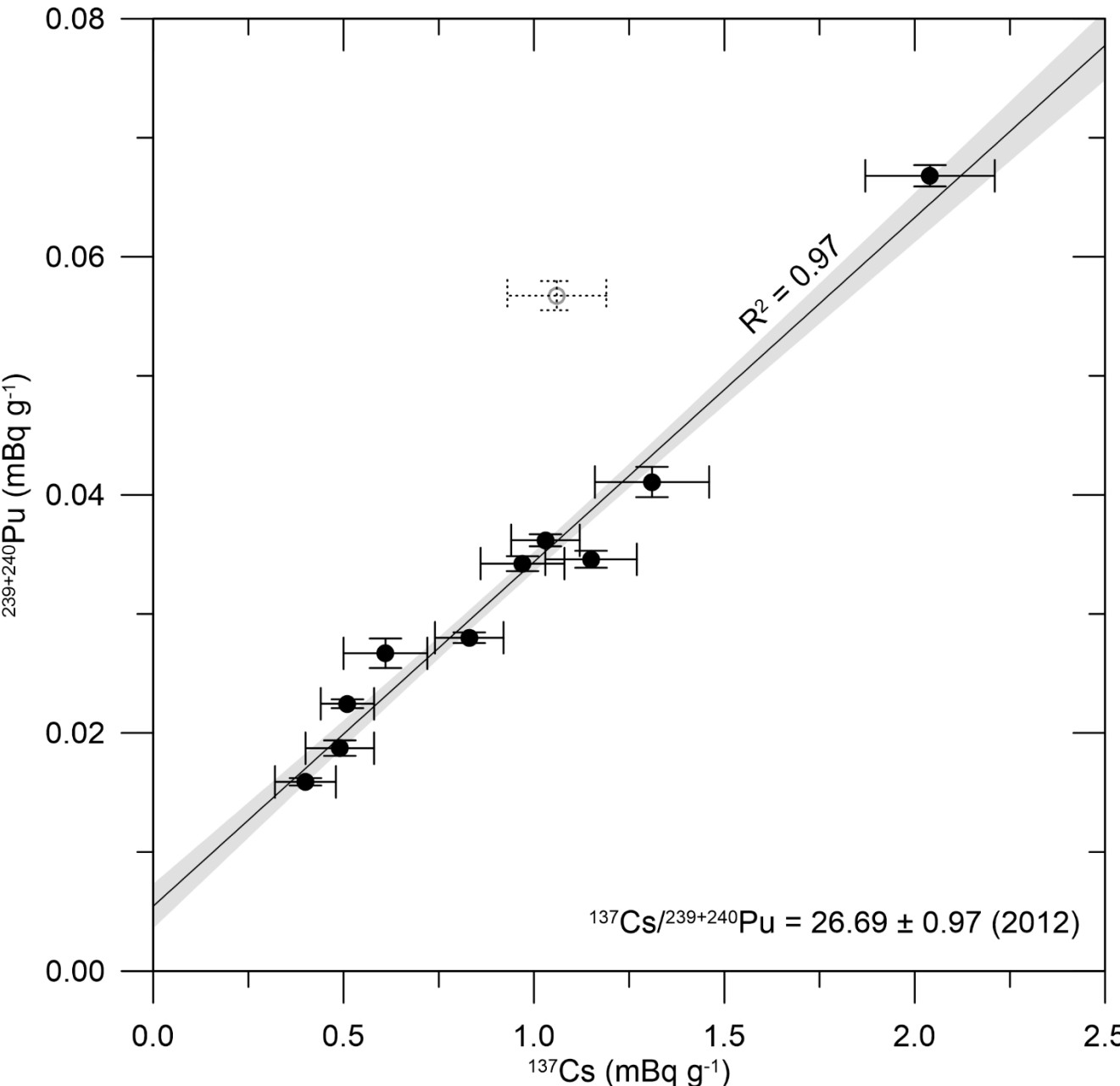

Figure 3: Correlation of [137]Cs and [239+240]Pu topsoil activities. [137]Cs data are shown with 1σ uncertainties (which equal the estimated measurement errors). [239+240]Pu activities were mostly measured in replicate, and the corresponding concentrations are weighted means and the uncertainties either dominated by AMS counting statistics (weighted mean error) or external sources of uncertainty (standard error). For single measurements, the 1σ measurement uncertainty provided by the AMS facilities dominates the final uncertainty. The majority of samples overlap with the linear regression (black line) and its 68% confidence interval (in grey). Sample HS0/0-20 has been excluded from the regression (greyed out; for discussion see text). The extrapolated regression intersects the ordinate at about 0.0050 mBq g⁻¹ (0.50 mBq kg⁻¹; unit conversion to mBq g⁻¹ due to lower level precision achieved by γ spectrometry). [137]Cs data have been decay-corrected to February 2012 (the time of measurement).

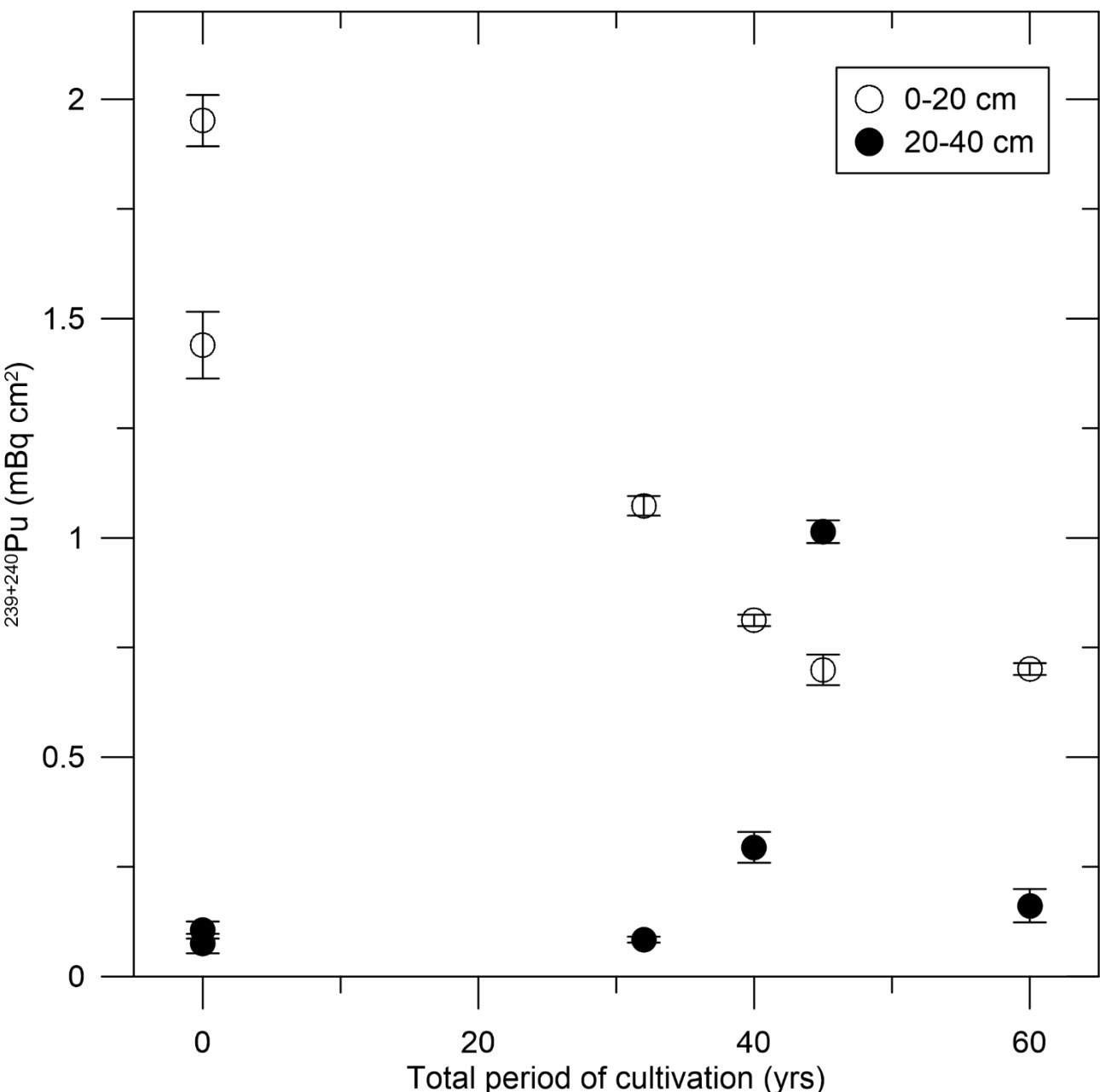

**Figure 4:** $^{239+240}$**Pu inventories at depth (20-40 cm) as compared to corresponding topsoil activities (0-20 cm). The analysis of plutonium activities at depth has been conducted for** $n = 6$ **samples, belonging to the agroecosystems Tweespruit (0, 32, 40, and 60 years of cultivation) and Harrismith (0 and 45 years of cultivation). At all sites but HS45, the nuclide concentration is significantly lower at depth than close to the surface. Error bars are 1σ uncertainties (see Fig. 2 for details).**




**Table 1: Agroecosystem details.**

|  | Harrismith | Kroonstad | Tweespruit |
|---|---|---|---|
| Abbreviation | HS | KR | TW |
| Number of plots investigated | 9 | 7 | 8 |
| Mean centre | 28.04°E, 28.38°S | 27.01°E, 27.90°S | 27.08°E, 29.24°S |
| Mean elevation | 1728 m a.s.l. | 1451 m a.s.l. | 1616 m a.s.l. |
| Mean distance between plots | 52.5 km | 14.4 km | 13.7 km |
| Climate region | Cwb | Cwa, Cwb | Cwb |
| Soil texture | Sandy loam | Loamy sand | Sandy loam |
| Ploughing depth | mostly 20 cm[b] | 20 cm | 20 cm |
| Soil type | Plinthic Lixisols/Typic Plinthustalfs | | |
| Crops (Average grain yields) | Maize, wheat, sunflower (2.2-3.8, 1.2-2.8, 2.2-3.8 t ha⁻¹) | | |

**The sampling sites were assigned to climate regions according to Beck et al. (2018). Cwa – Temperate, dry winter, hot summer; Cwb – temperate, dry winter, warm summer (in KR agroecosystem sample site KR57 only). The ploughing depth was reported to be 40 cm in HS30 and HS68. Soil types according to World Soil Reference Base (Iuss Working Group Wrb, 2015) and Soil Survey Staff (2014). Average grain yields valid for the time of sampling (1998), cited from Lobe et al. (2001).**






**Table 2: FRN inventories.**


| Ecotope | YOC (yr) | Depth (cm) | n | $^{239+240}$Pu (mBq cm$^2$) | | | (%) | | | $^{137}$Cs (mBq cm$^2$) | | |
|---|---|---|---|---|---|---|---|---|---|---|---|---|
| HS | 0 | 0-20 | 1 | 1.44 | ± | 0.08 | 100.0 | ± | 5.3 | 27.14 | ± | 4.24 |
| HS | 3.5 | 0-20 | 1 | 1.34 | ± | 0.04 | 93.3 | ± | 5.5 | - | | |
| HS | 5 | 0-20 | 2 | 1.28 | ± | 0.04 | 89.3 | ± | 5.3 | - | | |
| HS | 8 | 0-20 | 2 | 1.12 | ± | 0.02 | 77.6 | ± | 4.4 | - | | |
| HS | 10 | 0-20 | 2 | 0.75 | ± | 0.03 | 51.8 | ± | 3.5 | - | | |
| HS | 30 | 0-20 | 2 | 0.65 | ± | 0.01 | 45.1 | ± | 2.5 | - | | |
| HS | 45 | 0-20 | 1 | 0.70 | - | 0.03 | 48.6 | ± | 3.5 | 15.98 | ± | 2.91 |
| HS | 68 | 0-20 | 2 | 0.68 | ± | 0.05 | 47.5 | ± | 4.5 | - | | |
| HS | 90 | 0-20 | 2 | 0.72 | ± | 0.01 | 49.9 | ± | 2.8 | - | | |
| KR | 0 | 0-20 | 1 | 0.98 | ± | 0.03 | 100.0 | ± | 2.9 | 32.66 | ± | 3.62 |
| KR | 2.5 | 0-20 | 1 | 1.02 | ± | 0.03 | 103.8 | ± | 4.2 | - | | |
| KR | 20 | 0-20 | 2 | 0.65 | ± | 0.02 | 66.6 | ± | 2.5 | - | | |
| KR | 30 | 0-20 | 1 | 0.75 | ± | 0.03 | 77.1 | ± | 3.6 | - | | |
| KR | 40 | 0-20 | 1 | 0.48 | ± | 0.01 | 49.1 | ± | 1.9 | 12.08 | ± | 2.43 |
| KR | 57 | 0-20 | 2 | 0.68 | ± | 0.04 | 69.1 | ± | 4.9 | - | | |
| KR | 98 | 0-20 | 2 | 0.43 | ± | 0.01 | 44.1 | ± | 1.7 | - | | |
| TW | 0 | 0-20 | 3 | 1.95 | ± | 0.06 | 100.0 | ± | 3.0 | 59.57 | ± | 6.12 |
| TW | 8.0 | 0-20 | 3 | 1.23 | ± | 0.04 | 63.3 | ± | 2.7 | 39.30 | ± | 4.61 |
| TW | 12 | 0-20 | 3 | 1.08 | ± | 0.02 | 55.3 | ± | 1.9 | 30.69 | ± | 2.79 |
| TW | 22 | 0-20 | 1 | 1.01 | ± | 0.03 | 51.8 | ± | 2.2 | - | | |
| TW | 32 | 0-20 | 2 | 1.07 | ± | 0.02 | 55.0 | ± | 2.0 | 30.46 | ± | 3.54 |
| TW | 40 | 0-20 | 3 | 0.81 | ± | 0.01 | 41.6 | ± | 1.4 | 24.07 | ± | 2.68 |
| TW | 60 | 0-20 | 3 | 0.70 | ± | 0.01 | 35.9 | ± | 1.3 | 15.91 | ± | 2.22 |
| TW | 90 | 0-20 | 2 | 0.58 | ± | 0.02 | 29.7 | ± | 1.4 | 15.09 | ± | 2.80 |
| HS | 0 | 20-40 | 1 | 0.08 | ± | 0.02 | 5.2 | ± | 1.6 | - | | |
| HS | 45 | 20-40 | - | 1.01 | ± | 0.03 | 145.0 | ± | 8.1 | - | | |
| TW | 0 | 20-40 | 2 | 0.11 | ± | 0.02 | 5.4 | ± | 1.0 | - | | |
| TW | 32 | 20-40 | 2 | 0.08 | ± | 0.01 | 7.8 | ± | 0.7 | - | | |
| TW | 40 | 20-40 | - | 0.29 | ± | 0.04 | 36.3 | ± | 4.4 | - | | |
| TW | 60 | 20-40 | - | 0.16 | ± | 0.04 | 23.0 | ± | 5.5 | *below detection limit* | | |

Sample labelling as presented in the main text includes the abbreviation of the sample agroecosystem, years of cultivation (YOC) and sampling depth interval: HS - Harrismith, KR - Kroonstad, TW - Tweespruit. The number *n* of $^{239,240}$Pu replicate measurements includes both CologneAMS and ANU AMS measurement replicates. The quoted plutonium activities from replicate measurements are weighted means and the uncertainties either dominated by AMS counting statistics (weighted mean error) or external sources of uncertainty (standard error). For single measurements, the 1σ measurement uncertainty provided by the AMS facilities dominates the final uncertainty. Percentual activities of the topsoil samples are relative to the undisturbed reference sample for each agroecosystem. For the depth samples, the percentage values denote the difference against the corresponding topsoil samples. $^{137}$Cs data uncertainties equal 1σ measurement errors arising from μ spectrometry conducted at CSIRO. All $^{137}$Cs has been corrected for decay to February 2012.

