# Peer review of "Plutonium concentrations link soil organic matter decline to wind erosion in ploughed soils of South Africa"

_EGUsphere, 2024_

## Author Comment (AC1)

Pu manuscript – Reply to Anonymous Referee #1

Content

1. List of changes made to the manuscript due to a change in soil fraction considered (p. 1-3)
2. Updated figures and tables (p. 4-8)
3. Reply to Anonymous Referee #1 (p. 9-16)

**1. List of changes made to the manuscript due to a change in the soil fraction considered**

While discussing a reply to a comment of Referee #1 (AR1 #5), our team encountered a significant reporting mistake in how the physical processing of the samples was conducted. At some point in the past (the samples were already processed in 2012; see l. 134), a typo shifted the measured soil fraction from <2 mm to <20 µm. This mistake was then carried further, since it appeared a logical step to focus on this fraction. The truth is, however, that there was not enough original sample material left to separate the needed amount of <20 µm material (sandy soils). Thus, all measurements ($^{137}$Cs, $^{210}$Pbex, $^{239+240}$Pu) were conducted on the <2 mm fraction, i.e. the measurements include the bulk soil.

While this is a good example for the need of proper sample processing documentation, the issue brings a few changes to the data interpretation. Note, however, that the numbers as presented in the manuscript do either not change at all (measurements) or do not change significantly (correlations). An advantage is that we can now conveniently state bulk soil nuclide inventories instead of concentrations. In the following lines, we track the changes made to the manuscript arising from interpreting nuclide activities in the bulk soil, instead of the <20 µm fraction:

ll. 29-30: Updated for inventories: *Specifically, the original inventories of both $^{137}$Cs and $^{239+240}$Pu are approximately halved after ~20-40 years of cropping.*

l. 102: Updated to: *... possibly as a consequence of selective removal in this fraction and a relatively higher input of organic matter from crops, ...*

l. 171: Added "chemical": *"The chemical sample preparation for plutonium ..."*

l. 173: Rephrased to: *The physical preparation of the samples was conducted at the Institute of Crop Science and Resource Conservation, Bonn (sieving), and at the Commonwealth Scientific and Industrial Research Organisation, Land and Water Laboratories (CSIRO), Canberra (homogenisation).*

l. 174: Rephrased to: *In short, samples were sieved to obtain the <2 mm fraction and afterwards homogenised using a planetary mill.*

ll. 174-175: Sentence deleted, since we did not focus on the fraction <20 µm.

ll. 175-176: Rephrased to: *For AMS, about 20 g per sample were dried at 105°C to constant weight.*

ll. 176-177: Sentence deleted, since we did not focus on the fraction <20 µm.

ll. 195-197: Rephrased to: *To measure $^{137}$Cs, 50-70 g of the same homogenised material used for AMS were pressed into cylindrical counting discs to ensure a well-defined geometry.*

ll. 197-198: Rephrased to: *These sample measurements were conducted at CSIRO.*

ll. 225-227: Rephrased to: *From these $^{239+240}$Pu activities per mass (here also termed "specific activities") we derived inventories, i.e. activities per area, by including sampling depth and bulk density data (Table S1).*

ll. 240-241: Updated for inventories.

l. 258: Updated for bulk soil.

ll. 259-260: Updated for inventories: *The measured inventories in the top 20 cm of soil span a wide range between 0.43 ± 0.01 mBq cm² (KR98/0-20) and 1.95 ± 0.06 mBq cm² (TW0/0-20).*

ll. 261-263: Updated for inventories: *Similarly, the other samples from the uncultivated plots in the other two agroecosystems also have the largest inventories in their respective agroecosystems (HS0/0-20 1.44 ± 0.08 mBq cm²; KR0/0-20 0.98 ± 0.03 mBq cm²).*

ll. 264-270: Updated for inventories *(R² = 0.76).*

l. 270: Sentence added: *Sample KR2.5/0-20 shows an elevated relative inventory of 103.84 ± 4.22% (relative concentration 99.93 ± 3.00%) but does overlap within uncertainties with the defined initial activity. Hence, the sample was excluded from the fit.*

ll. 270-271: Updated for inventories: *From the fit, $I_{eq}$ equals 56.03 ± 6.01% ($1\sigma_x$), and $\tau$ equals 6.86 ± 3.03 years.*

l. 276: Deleted: *... in the <20 μm fraction ...*

l. 284: Updated for inventories.

ll. 287-291: Updated for inventories: *The results indicate that inventories are generally much lower than in the top 20 cm, ranging from ~5 to 36% of what is measured in the corresponding topsoil sample (Table 2; Fig. 5). Sample HS45/20-40 is, however, a conspicuous exception with a surprisingly high inventory of 1.01 ± 0.03 mBq cm² in the 20-40 cm interval, which is even higher than the 0.70 ± 0.03 mBq cm² measured in the uppermost 20 cm of the soil (HS45/0-20).*

l. 294: Updated for bulk soil.

l. 301: Sentence added: *In line with this argumentation, the $^{239+240}$Pu inventories obtained from the native grassland composite samples are in the range expected for surface samples located within 20-30°S, which has been constrained to be 1.44 ± 0.59 mBq cm⁻² (Hardy et al. 1973).*

[revised manuscript text omitted]

**Figure S1: Linear correlations between N contents (Lobe et al., 2001) and $^{239+240}$Pu concentrations in the bulk soil. Most**
**plutonium samples depict replicate measurements; the corresponding concentrations are weighted means and the**
**uncertainties either dominated by AMS counting statistics (weighted mean error) or external sources of uncertainty**
**(standard error). For single measurements, the 1σ measurement uncertainty provided by the AMS facilities dominates**
**the final uncertainty. (vertical error bars either represent 1σ from the mean of several replicates or a 1σ error-**
**propagated uncertainty dominated by the AMS measurement uncertainty). Samples from sites that have been cropped**
**before 1963 were excluded from the regression (greyed out data points).**

[Figure]

Figure S2: Silt fraction as measured by Lobe et al. (2001) and Amelung et al. (2002). Uncertainties are 1σ standard
**deviations of the arithmetic means ($n \geq 2$ replicates per sample).**

**Table. 2: FRN inventories.**

| Ecotope | YOC (yr) | Depth (cm) | n | $^{239+240}$Pu (mBq cm$^2$) | | | $^{239+240}$Pu (%) | | | $^{137}$Cs (mBq cm$^2$) | | |
|---|---|---|---|---|---|---|---|---|---|---|---|---|
| HS | 0 | 0-20 | 1 | 1.44 | ± | 0.08 | 100.0 | ± | 5.3 | 27.14 | ± | 4.24 |
| HS | 3.5 | 0-20 | 1 | 1.34 | ± | 0.04 | 93.3 | ± | 5.5 | - | | |
| HS | 5 | 0-20 | 2 | 1.28 | ± | 0.04 | 89.3 | ± | 5.3 | - | | |
| HS | 8 | 0-20 | 2 | 1.12 | ± | 0.02 | 77.6 | ± | 4.4 | - | | |
| HS | 10 | 0-20 | 2 | 0.75 | ± | 0.03 | 51.8 | ± | 3.5 | - | | |
| HS | 30 | 0-20 | 2 | 0.65 | ± | 0.01 | 45.1 | ± | 2.5 | - | | |
| HS | 45 | 0-20 | 1 | 0.70 | - | 0.03 | 48.6 | ± | 3.5 | 15.98 | ± | 2.91 |
| HS | 68 | 0-20 | 2 | 0.68 | ± | 0.05 | 47.5 | ± | 4.5 | - | | |
| HS | 90 | 0-20 | 2 | 0.72 | ± | 0.01 | 49.9 | ± | 2.8 | - | | |
| KR | 0 | 0-20 | 1 | 0.98 | ± | 0.03 | 100.0 | ± | 2.9 | 32.66 | ± | 3.62 |
| KR | 2.5 | 0-20 | 1 | 1.02 | ± | 0.03 | 103.8 | ± | 4.2 | - | | |
| KR | 20 | 0-20 | 2 | 0.65 | ± | 0.02 | 66.6 | ± | 2.5 | - | | |
| KR | 30 | 0-20 | 1 | 0.75 | ± | 0.03 | 77.1 | ± | 3.6 | - | | |
| KR | 40 | 0-20 | 1 | 0.48 | ± | 0.01 | 49.1 | ± | 1.9 | 12.08 | ± | 2.43 |
| KR | 57 | 0-20 | 2 | 0.68 | ± | 0.04 | 69.1 | ± | 4.9 | - | | |
| KR | 98 | 0-20 | 2 | 0.43 | ± | 0.01 | 44.1 | ± | 1.7 | - | | |
| TW | 0 | 0-20 | 3 | 1.95 | ± | 0.06 | 100.0 | ± | 3.0 | 59.57 | ± | 6.12 |
| TW | 8.0 | 0-20 | 3 | 1.23 | ± | 0.04 | 63.3 | ± | 2.7 | 39.30 | ± | 4.61 |
| TW | 12 | 0-20 | 3 | 1.08 | ± | 0.02 | 55.3 | ± | 1.9 | 30.69 | ± | 2.79 |
| TW | 22 | 0-20 | 1 | 1.01 | ± | 0.03 | 51.8 | ± | 2.2 | - | | |
| TW | 32 | 0-20 | 2 | 1.07 | ± | 0.02 | 55.0 | ± | 2.0 | 30.46 | ± | 3.54 |
| TW | 40 | 0-20 | 3 | 0.81 | ± | 0.01 | 41.6 | ± | 1.4 | 24.07 | ± | 2.68 |
| TW | 60 | 0-20 | 3 | 0.70 | ± | 0.01 | 35.9 | ± | 1.3 | 15.91 | ± | 2.22 |
| TW | 90 | 0-20 | 2 | 0.58 | ± | 0.02 | 29.7 | ± | 1.4 | 15.09 | ± | 2.80 |
| HS | 0 | 20-40 | 1 | 0.08 | ± | 0.02 | 5.2 | ± | 1.6 | - | | |
| HS | 45 | 20-40 | - | 1.01 | ± | 0.03 | 145.0 | ± | 8.1 | - | | |
| TW | 0 | 20-40 | 2 | 0.11 | ± | 0.02 | 5.4 | ± | 1.0 | - | | |
| TW | 32 | 20-40 | 2 | 0.08 | ± | 0.01 | 7.8 | ± | 0.7 | - | | |
| TW | 40 | 20-40 | - | 0.29 | ± | 0.04 | 36.3 | ± | 4.4 | - | | |
| TW | 60 | 20-40 | - | 0.16 | ± | 0.04 | 23.0 | ± | 5.5 | *below detection limit* | | |

**Sample labelling as presented in the main text includes the abbreviation of the sample agroecosystem, years of**
**cultivation (YOC) and sampling depth interval: HS - Harrismith, KR - Kroonstad, TW - Tweespruit. The number *n* of**

**²³⁹,²⁴⁰Pu replicate measurements includes both CologneAMS and ANU AMS measurement replicates. The quoted**
**plutonium activities from replicate measurements are weighted means and the uncertainties either dominated by AMS**
**counting statistics (weighted mean error) or external sources of uncertainty (standard error). For single measurements,**
**the 1σ measurement uncertainty provided by the AMS facilities dominates the final uncertainty. Percentual activities**
**of the topsoil samples are relative to the undisturbed reference sample for each agroecosystem. For the depth samples,**
**the percentage values denote the difference against the corresponding topsoil samples. ¹³⁷Cs data uncertainties equal 1σ**
**measurement errors arising from μ spectrometry conducted at CSIRO. All ¹³⁷Cs has been corrected for decay to**
**February 2012.**

**3. Reply to Anonymous Referee #1**

Dear Anonymous Referee #1, thank you for reviewing our manuscript and for providing your detailed assessment. We address the points you raised below.

**AR1 #1: The serious scientific challenge to this study is the use of Pu and SOM concentrations rather than mass balance, since the latter is required to demonstrate mass redistribution as typically laid out by erosion studies based on fallout radionuclides (e.g. authors He, Walling, Wallbrink, Mabit, Alewell, Meusberger etc.). I recognize that the present authors are limited by the work of their predecessors but nonetheless this issue should require more direct and explicit treatment here. Else it sounds as though the attribution of declines in Pu and SOM concentrations with cultivation history to wind erosion is fait accompli, while there is otherwise no direct evidence of the process of wind erosion per se presented here.**

Reply AR1 #1: We agree and add the following information and statements:

l. 105: *About 100 km to the northwest of the Tweespruit sites, Wiggs and Holmes (2011) measured dust fluxes on a flat (<2°) ploughed field belonging to the Grasslands farm near Bloemfontain. The authors reported a total dust deposition of 48.19 g cm$^{-2}$ (0.48 g m$^{-2}$ day$^{-1}$) from the local, wind-eroding sandy soils for a timespan ranging 99 days between August and November 2007. For the sites we investigate in our study, a re-assessment of the silt fraction content [partially unpublished, measured by Lobe et al. (2001) and Amelung et al. (2002); Tab. S1] reveals a linear increase in south-eastern direction (R2 = 0.73; Fig. S1), which follows the general trajectories of dust plumes in South Africa (Eckard et al. 2020).*

l. 170: *A clear disadvantage of the applied sampling scheme is the lack of high-resolution depth profile samples, which was not required for the originally intended sample analyses. As a consequence, we are unable to present FRN mass depth profile data, and thus cannot reasonably infer mass redistribution rates as typically presented in FRN studies (e.g. Alewell et al. 2014, Lal et al. 2013, Meusburger et al. 2018).*

l. 423: rephrased to: *To resolve this issue, longer timescales need to be monitored and high-resolution depth profiles sampling implemented in future studies.*

l. 426: updated to: *In our study, we measure the lowest activities in the upwind sites. The composite grassland sample HS0/0-20 has an activity twice as high as the most north-west located sites (KR0/0-20), generally coinciding with SOM patterns as published by Lobe et al. (2001). This pattern is further reflected by grain size data, indicating an increase in the silt fraction towards the south-east (Fig. S1). Consequently, KR soils had the lowest silt fraction of all soils (Fig. S2) by the time of sampling.*

**AR1 #2: There are alternative explanations for a change in SOM and Pu concentrations that may be challenged more directly by the authors. First, the foremost influence on SOM and Pu concentrations in soil upon first tilling will be the tilling itself, since in native soils the highest concentrations of both are at the soil surface. If the tilling process is anything but perfectly homogenizing in the 0-20 cm soil, and there were any bias in sampling depth relative to tilling depth (say, 20 cm and 30 cm respectively), would the appearance of the concentrations over time not be exactly what we see in Figure 2? I wonder what assurance the authors provide that the observed patterns are simply not an artifact of tilling and sampling?**

Reply AR1 #2: We interpret your comment as follows (please correct us, if we misunderstood): In native grassland, SOM and Pu will be stored in the uppermost part of the soil column. Then, the soil is ploughed, relocating SOM/Pu and shifting the peak concentrations down the profile, i.e. not completely homogenising it. If we then take our sample, and the sampling depth does not capture the ploughing horizon entirely, we would miss a certain fraction of Pu/SOM. This fraction is than brought up again
after more ploughing and incorporated to the range that we sample. By always missing the lower part of
the inhomogeneous ploughing horizon we would face a continuous in- or outflux of Pu/SOM over time.
So, basically, there are three issues to be addressed: homogenisation, ploughing depth and sampling
depth. While we cannot rule out ploughing depth as a source of uncertainty, which we stated in l. 159,
we focus on sampling depth and, more importantly, homogenisation.

Information on homogenisation added to rewritten Section 1.3 (see reply to AR2 #10): *Vertical*
*migration in the soil column can also be achieved due to physical processes, such as bioturbation or*
*tillage. The latter, which is of special importance to this study, has been shown to homogenise FRN*
*concentrations throughout the $A_p$ horizon rapidly, e.g. after ~1-4 times of soil inversion (Schimmack et*
*al. 1994, Hoshino et al. 2015).*

l. 162: Information about sampling added: *Five subsamples, taken by using a steel cylinder, were*
*amalgamated per plot to obtain the final sample.*

l. 321: Error propagation (which will of course not affect the depth vs. homogenisation issue) and
information about (accurate) sampling added: *However, the slight variation in soil densities resulted in*
*larger uncertainties of the pooled reference samples due to error propagation, when compared to the*
*ploughed plots. An accurate sampling depth was achieved by using a 20-cm long steel cylinder for*
*sampling, but we propagate a depth error of 0.5 cm for calculating the inventories.*

ll. 381-391: Updated and rephrased to: *Bulk SOC has been shown to approach an equilibrium*
*concentration of 39.4 ± 2.0% of the initial values after 33.6 ± 8.0 years of cropping, with $\tau = 6.7 ± 1.7$*
*years (Fig. 2B; Lobe et al., 2001; their exponential model). The SOC equilibrium concentration differs*
*from the Pu inventory equilibrium (and, likewise, the Pu concentration equilibrium, Fig. S5). This*
*deviation may to a certain extent be explained by the cut-off of Pu data included in the fit after 35 years*
*of cropping (Sect. 4.3). On the other hand, the similarity in time constants might indicate that the*
*plutonium was similarly distributed as SOC in the upper 20 cm of the soil soon after initial ploughing,*
*as homogenisation appears to be generally achieved rapidly (Sect. 2.1). Consequently, SOM contents*
*and Pu activities both reflect largest rates of decline during the first years after native grasslands were*
*converted to arable land. This similar behaviour of $^{239+240}Pu$ activities and SOM content over time*
*indicates a strong linkage between both variables. The relationship is underscored by high correlation*
*coefficients ($R^2 = 0.56$ to 0.99; Fig. 2C) between the plutonium concentrations (1963-1998) and bulk*
*soil SOC contents as measured by Lobe et al. (2001) (Table S5). Similar $R^2$ values (0.63 – 0.99) are*
*obtained when $^{239+240}Pu$ is correlated to total N contents (Fig. S1).*

**AR1 #3: Next, the authors should acknowledge that the lower boundary of the 0-20 cm soil depth**
**is not closed. They do point this out with respect to one location that actually as higher Pu in 20-**
**40 cm than 0-20 cm, but this is attributed to deeper tilling. While this actually speaks to my point**
**(1) above, I also highlight the potential for leakage over time of both SOM and Pu below the 20**
**cm boundary. The normal migration of organometallic complex at just 0.2 mm y-1 could export**
**1% of Pu and SOC to deeper soil, for example. Compounded with time this could easily explain**
**the scale of Pu and SOC loss over decades.**

**However, it is interesting to observe that the two pulses have different time zeroes. Any SOC pulse**
**from inherited O-horizon is necessarily timed with onset of cultivation, whereas the Pu pulse is**
**independent of that. In fact the Pu pulse is centered at a cultivation time of approximately 40 years**
**... and there appears no obvious pattern in the data related to this (excepting the noted site with**
**higher Pu >20 cm which I surmised is related to deep tilling near in time to peak Pu deposition).**
**That the SOC and Pu patterns both reflect time since cultivation would seem therefore to be good**

**corroboration that it is some extant property of the soil that is regulating retention of SOC and Pu that are introduced at the soil surface rather than any artifact of tilling.**

Reply to AR1 #3: We tried to address the leakage issue by measuring Plutonium in a selection of depth samples (Fig. 5). However, we did not put much emphasis on the two samples that indicate elevated inventories (> 10% w.r.t. the topsoil sample) so far.

l. 236: peak plutonium deposition added and rephrased: *Since our sampling strategy included a spatial averaging of sampling material from each plot investigated (Sect. 2.2), the best explanation for the elevated plutonium activity in HS45/20-40 may be related to a former ploughing to 40 cm near in time to peak plutonium deposition that was not recorded during farmers' interviews or to sample contamination. Elevated inventories measured in two further depth samples might point to a certain degree of leakage of plutonium towards greater depths, but not necessarily in the pre-fallout soils (Sect. 4.3). We also note that in a reasonably comparable setting (Bsh climate; Big Spring, USA), $^{137}Cs$ and $^{239+240}Pu$ concentrations dropped sharply below the $A_p$ horizon in an Aridic Paleustalf that had been cultivated since 1915 (Van Pelt et al., 2007; Van Pelt and Ketterer, 2013).*

l. 371: leakage added: *Likewise, a possible incorporation of Pu-marked plant material into the soil column after harvesting might have contributed to elevated inventories found in the three depth samples with cropping histories exceeding 35 years (HS45/20-40, TW40/20-40, and TW60/20-40). In case of wet fallout deposition, such an enhanced downward migration could also have been promoted by the physical disturbance of the ploughed soil (cf. Das Gupta et al., 2006). These factors appear not to have been significant for the plot that was converted after the peak episode of global fallout (TW32/20-40). However, the topsoil samples TW40/0-20 and TW60/0-20 had the strongest inventory losses of all samples (Fig. 2a; Table 2), and show the highest negative residues against the exponential model. Thus, and given the general low scatter of the post-35 YOC data points, we may argue that if significant migration of Pu-marked soil particles below 0-20 cm has occurred, the two samples in question could represent the cases of maximum leakage in our dataset.*

l. 418: leakage added: *Furthermore, a certain degree of leakage of plutonium-marked particles to greater depths could contribute to a lowering of $^{239+240}Pu$ activities in the topsoil over time.*

**AR1 #4: It should also be noted that SOC and Pu differ in that there are continuous SOC inputs at the soil surface through ongoing plant growth, whereas there are no ongoing Pu inputs with possible exception of relatively minor remobilization through erosive process. I wonder if this difference explains the different long-term trajectories of SOC and Pu in Figure 2 ... Pu continues to decline due possibly to leakage out the bottom (point above), whereas SOC is at some steady state with respect to ongoing inputs.**

Reply to AR1 #4: We agree that there are different inputs, but emphasise that previous studies also used biexponential models to capture a continuous decrease in SOM contents in the soils. Lobe et al. (2005) address the SOC in- and outputs. While there is a replacement in old SOC with new SOC (65% of SOC is crop-derived after 90 years, on average), in- and output rates are very similar with slightly higher rates for the outputs (equilibria after ~20 yrs). We find your thoughts exciting but point to the lack of causality for the topsoil Pu data exceeding 35 years. However, we include the leakage hypothesis as presented for AR1 #3 (l. 418). We also refer to our reply AR1 #7.

**AR1 #5: Finally, the authors should make clear in this paper (and not through references), the handling of the samples to produce the <20 um fraction, and especially what % this represents of the whole sample. Otherwise it remains unclear how the <20 um fraction might relate to soil mass balance, and whether a decrease in SOC and Pu concentrations in this fraction is truly attributable**

to mass balance, or alternatively to a change in texture or other soil properties that regulate
carbon and fallout metal retention. More details on the <20 um fraction are especially important
since this fraction may now be interpreted as the 'mineral associated organic carbon' or MOAC
fraction, which the author touch on tangentially as possibly related to changes in carbon
sequestration over time.

Reply to AR1 #5: As detailed above, this comment revealed our wrong sample processing recordings
(or, to be more precise, how these records were interpreted after a long time of hibernation). We thank
you very much for pushing us there!

**AR1 #6: In arguing for wind erosion to the authors do conclude with some observations on a**
**possible gradient in Pu concentrations along wind fetch. If this were the case, could this also be**
**correlated with changes in soil texture or % fines due to deflation? Such a correlation would make**
**a fine figure and would provide some independent evidence for wind erosion that is otherwise**
**lacking.**

Reply to AR1 #6: True! We have calculated the distance of our sampling sites along a NW-SE transect
(distance = 1 km at the most northwesternmost sampling site) to obtain a correlation (see also new lines
of text as presented in our reply to AR1 #1):

[Figure]

**Figure S2: Soil fraction 2-20 μm and distance in south-eastern direction of individual samples (pooled samples**
**excluded). The grain size data has been measured by Lobe et al. (2001) and Amelung et al. (2002) (partially**
**unpublished).**

**AR1 #7: Finally, the peculiar problem of a Pu point source some 40 years prior to soil sampling**
**might be countered if the authors were able to use Pb-210 which was likely measured concurrently**
**with their Cs-137 measurements. Similar to Cs it may be that the Pb-210 half-life precluded robust**
**measurement from old archive samples, but even in this case it would be worth including a**

**statement to this effect since otherwise Pb-210 could be quite valuable to the study. It would**
**powerful to show for example that Pb-210 more precisely mirrors SOC due to continuous input**
**to both through the history of the experiment ... or not!**

Reply to AR1 #7: Indeed, $^{210}Pb_{ex}$ has been measured. We did not include it before because of large error
ranges, but include it now:

l. 477: *An approach to overcome the problem of a point source in time determined by anthropogenic*
*global fallout involves the quantification of excess $^{210}Pb$, hereafter referred to as $^{210}Pb_{ex}$. This naturally*
*occurring FRN has been widely used for erosion studies elsewhere (e.g. Matisoff, 2014; Mabit et al.,*
*2008; Hu and Zhang, 2019; Meusburger et al., 2018), and was measured at CSIRO alongside $^{137}Cs$*
*(Tab. S6). From this data, general trends of concurrently decreasing $^{210}Pb_{ex}$ activities with decreasing*
*$^{137}Cs$ activities (Fig. S3) and SOC content (Fig. S4) might be inferred. However, the short half-life (~22.2*
*yrs) of the nuclide plus >10 years of sample storage and generally low environmental concentrations*
*cause propagated measurement errors of 16-30% of the respective means. Hence, we refrain from*
*providing a detailed interpretation of $^{210}Pb_{ex}$ activities in our samples.*

[Figure]

**Figure S3: $^{137}Cs$ and $^{210}Pb_{ex}$ topsoil activities. All data shown with 1σ uncertainties, which equal the estimated**
**measurement errors for $^{137}Cs$ and include error propagation for the calculation of $^{210}Pb_{ex}$. Linear fits are shown for**
**visual comparison of the <35 YOC (years of cultivation) and >35 YOC samples, respectively (dashed lines).**

[Figure]

**Figure S2: Soil organic carbon (SOC) contents and $^{210}Pb_{ex}$ activities in the bulk topsoil. $^{210}Pb_{ex}$ data includes error-**
**propagated 1σ measurement uncertainties.**

**Table S6: $^{210}Pb_{ex}$ specific activies.**

| Ecotope | YOC (yr) | Depth (cm) | $^{210}Pb_{ex}$ (Bq kg$^{-1}$) | | |
|---------|----------|------------|------|---|------|
| HS | 0 | 0-20 | 7.80 | ± | 2.05 |
| HS | 45 | 0-20 | 6.10 | ± | 1.88 |
| KR | 0 | 0-20 | 10.44 | ± | 1.70 |
| KR | 40 | 0-20 | 5.67 | ± | 1.36 |
| TW | 0 | 0-20 | 12.27 | ± | 2.53 |
| TW | 8.0 | 0-20 | 8.76 | ± | 2.13 |
| TW | 12 | 0-20 | 9.58 | ± | 1.80 |
| TW | 32 | 0-20 | 9.15 | ± | 1.88 |
| TW | 40 | 0-20 | 9.37 | ± | 1.77 |
| TW | 60 | 0-20 | 9.31 | ± | 1.55 |
| TW | 90 | 0-20 | 8.28 | ± | 1.77 |
| TW | 60 | 20-40 | *below detection limit* | | |

$^{210}Pb_{ex}$ was measured alongside $^{137}Cs$. Measurement procedure as e.g. detailed in Swarzenski (2014). Equilibrium $^{222}Rn$
was calculated from the weighted average of $^{214}Pb$ (295 and 352 keV) and $^{214}Bi$ (609 keV). To calculate final $^{210}Pb_{ex}$
activities, $^{222}Rn$ activities were subtracted from $^{210}Pb$ (46.5 keV) activities.

YOC: Years of cultivation.

**AR1 #8: c.r. line 426: for additional evidence for wind erosion, examine residuals from exponential**
**fits of SOM and Pu along wind direction ... do concentrations (residuals) for both SOM and Pu**
**covary tightly?**

Reply to AR1 #8: Assessing the residuals reveals no strong covariation – SOM has a minor tendency to positive residuals (measured value – modelled value) with increasing distance, while Pu shows a stronger tendency to decreasing values (the latter for the <35 years data only; linear R2 = 0.47, 0 YOC excluded). However, as we point out in ll. 419-422, there is no real certainty about the model type, and we feel that trying more exact fitting would be an over-elaboration given the limited dataset.

**AR1 #9: line 346: please clarify, aggregate size increases with time since cultivation?**

Reply to AR 1 #9: Indeed, the aggregate mass >250 increases (relative to total soil mass) with time since cultivation.

**AR1 #10: line 384: excellent point**

Reply to AR 1#10: Very sad to see that the reasoning you liked had to be changed due to the wrong soil fraction used by us. See our new line of argumentation for ll. 381-391 above.

**AR1 #11: Figure 2: both Pu (panel a) and SOM (panel b) show exponential declines with cultivation time, but the dropoff appears much more steep for Pu than for SOC. I wonder about the significance of this, or is it simply a result of intersite variability? It may be due to different pulse input histories?**

Reply to AR 1#11: Similar to reply AR 1#10 – Time constants have changed now. We apologise for the extra efforts.

**AR1 #12: Figure 3: this is a methods figure and I would recommend placing in supplemental materials, or simply omitting it while replacing with relevant summary statistics in the methods section.**

Reply to AR1 #12: We have moved the figure to the supplement.

**AR1 #13: Figure 4: it would be nice to indicate the Pu:Cs fallout ratio decay-corrected to 2012. The correlation here is impressive, but this does not necessarily indicate that both Pu and Cs are retained to same degree, only that the fraction that IS retained is retained to same degree. This occurs in lake sediments for example where Pu and Cs are similarly strongly correlated with depth, but by mass balance as much as half of Cs depositional flux is missing presumably due to higher solubility in the water column.**

Reply to AR1 #13: We have added the Cs/Pu fallout ratio decay-corrected to 2012 (26.69 ± 0.97) to Figure 4/3. We added the following piece of text:

l. 314: *However, any leakage could equally affect the isotope concentrations and may thus not be reflected by the ratio.*

**AR1 #14: Figure 5: high Pu in 40-50 year cultivated sites... did first plowing quickly follow period of peak Pu deposition, and mix Pu into subsurface 20-40 cm soils thereby minimizing susceptibility to erosion? What would this mean for mass balance and assumption that Pu is lost to wind erosion at the soil surface?**

Reply to AR1 #14: We might speculate that the incorporation of Pu-marked plant material into the soil
and/or the physical disruption of the soil during fallout could contribute to this effect. However, we also
need to acknowledge that the two other sites that show elevated Pu inventories at depth have the highest
decrease of all topsoil samples (happy coincidence). See our reply to AR1 #3.

---

## Author Comment (AC2)

Pu manuscript – Reply to Anonymous Referee #2

Content

**1. List of changes made to the manuscript due to a change in the soil fraction considered**

While discussing a response to a comment of Referee #1 (AR1 #5), our team discovered a significant reporting error in the physical processing of the samples. At some point in the past (the samples were already processed in 2012; see l. 134), a typo shifted the measured soil fraction from <2 mm to <20 µm. This mistake was then carried further, since it appeared a logical step to focus on this fraction (higher concentration of FRNs in that fraction). However, it turned out that there was not enough original sample material left to separate the required amount of the <20 µm fraction (sandy soils). As a result, all measurements ($^{137}$Cs, $^{210}$Pbex, $^{239+240}$Pu) were actually conducted on the <2 mm fraction, i.e. the measurements include the bulk soil.

This discovery highlights the importance proper sample processing documentation and brings a few changes to the data interpretation. It is worth noting that the numbers as presented in the manuscript do either not change at all (measurements) or do not change significantly (correlations). An advantage is that we can now report bulk soil nuclide inventories instead of concentrations. In the following section, we track the adjustments made to the manuscript as a result of interpreting nuclide activities in the bulk soil rather than in the <20 µm fraction:

ll. 29-30: Updated for inventories: *Specifically, the original inventories of both $^{137}$Cs and $^{239+240}$Pu are approximately halved after ~20-40 years of cropping.*

l. 102: Updated to: *... possibly as a consequence of selective removal in this fraction and a relatively higher input of organic matter from crops, ...*

l. 171: Added "chemical": *"The chemical sample preparation for plutonium ..."*

l. 173: Rephrased to: *The physical preparation of the samples was conducted at the Institute of Crop Science and Resource Conservation, Bonn (sieving), and at the Commonwealth Scientific and Industrial Research Organisation, Land and Water Laboratories (CSIRO), Canberra (homogenisation).*

l. 174: Rephrased to: *In short, samples were sieved to obtain the <2 mm fraction and afterwards homogenised using a planetary mill.*

ll. 174-175: Sentence deleted, since we did not focus on the fraction <20 µm.

ll. 175-176: Rephrased to: *For AMS, about 20 g per sample were dried at 105°C to constant weight.*

ll. 176-177: Sentence deleted, since we did not focus on the fraction <20 µm.

ll. 195-197: Rephrased to: *To measure $^{137}$Cs, 50-70 g of the same homogenised material used for AMS were pressed into cylindrical counting discs to ensure a well-defined geometry.*

ll. 197-198: Rephrased to: *These sample measurements were conducted at CSIRO.*

ll. 225-227: Rephrased to: *From these $^{239+240}$Pu activities per mass (here also termed "specific activities") we derived inventories, i.e. activities per area, by including sampling depth and bulk density data (Table S1).*

ll. 240-241: Updated for inventories.

l. 258: Updated for bulk soil.

ll. 259-260: Updated for inventories: *The measured inventories in the top 20 cm of soil span a wide*
*range between $0.43 \pm 0.01$ mBq cm$^2$ (KR98/0-20) and $1.95 \pm 0.06$ mBq cm$^2$ (TW0/0-20).*

ll. 261-263: Updated for inventories: *Similarly, the other samples from the uncultivated plots in the other*
*two agroecosystems also have the largest inventories in their respective agroecosystems (HS0/0-20 1.44*
*$\pm 0.08$ mBq cm$^2$; KR0/0-20 $0.98 \pm 0.03$ mBq cm$^2$).*

ll. 264-270: Updated for inventories *($R^2 = 0.76$).*

l. 270: Sentence added: *Sample KR2.5/0-20 shows an elevated relative inventory of $103.84 \pm 4.22\%$*
*(relative concentration $99.93 \pm 3.00\%$) but does overlap within uncertainties with the defined initial*
*activity. Hence, the sample was excluded from the fit.*

ll. 270-271: Updated for inventories: *From the fit, $I_{eq}$ equals $56.03 \pm 6.01\%$ ($1\sigma_x$), and $\tau$ equals $6.86 \pm$*
*3.03 years.*

l. 276: Deleted: *... in the <20 µm fraction ...*

l. 284: Updated for inventories.

ll. 287-291: Updated for inventories: *The results indicate that inventories are generally much lower than*
*in the top 20 cm, ranging from ~5 to 36% of what is measured in the corresponding topsoil sample*
*(Table 2; Fig. 5). Sample HS45/20-40 is, however, a conspicuous exception with a surprisingly high*
*inventory of $1.01 \pm 0.03$ mBq cm$^2$ in the 20-40 cm interval, which is even higher than the $0.70 \pm 0.03$*
*mBq cm$^2$ measured in the uppermost 20 cm of the soil (HS45/0-20).*

l. 294: Updated for bulk soil.

l. 301: Sentence added: *In line with this argumentation, the $^{239+240}$Pu inventories obtained from the native*
*grassland composite samples are in the range expected for surface samples located within 20-30°S,*
*which has been constrained to be $1.44 \pm 0.59$ mBq cm$^{-2}$ (Hardy et al. 1973).*

[revised manuscript text omitted]

**Figure S1: Linear correlations between N contents (Lobe et al., 2001) and $^{239+240}$Pu concentrations in the bulk soil. Most**
**plutonium samples depict replicate measurements; the corresponding concentrations are weighted means and the**
**uncertainties either dominated by AMS counting statistics (weighted mean error) or external sources of uncertainty**
**(standard error). For single measurements, the 1σ measurement uncertainty provided by the AMS facilities dominates**
**the final uncertainty. (vertical error bars either represent 1σ from the mean of several replicates or a 1σ error-**
**propagated uncertainty dominated by the AMS measurement uncertainty). Samples from sites that have been cropped**
**before 1963 were excluded from the regression (greyed out data points).**

[Figure]

**Figure S2: Silt fraction as measured by Lobe et al. (2001) and Amelung et al. (2002). Uncertainties are 1σ standard**
**deviations of the arithmetic means ($n \geq 2$ replicates per sample).**

**Table. 2: FRN inventories.**

| Ecotope | YOC (yr) | Depth (cm) | n | $^{239+240}$Pu (mBq cm$^2$) | | | $^{239+240}$Pu (%) | | | $^{137}$Cs (mBq cm$^2$) | | |
|---|---|---|---|---|---|---|---|---|---|---|---|---|
| HS | 0 | 0-20 | 1 | 1.44 | ± | 0.08 | 100.0 | ± | 5.3 | 27.14 | ± | 4.24 |
| HS | 3.5 | 0-20 | 1 | 1.34 | ± | 0.04 | 93.3 | ± | 5.5 | - | | |
| HS | 5 | 0-20 | 2 | 1.28 | ± | 0.04 | 89.3 | ± | 5.3 | - | | |
| HS | 8 | 0-20 | 2 | 1.12 | ± | 0.02 | 77.6 | ± | 4.4 | - | | |
| HS | 10 | 0-20 | 2 | 0.75 | ± | 0.03 | 51.8 | ± | 3.5 | - | | |
| HS | 30 | 0-20 | 2 | 0.65 | ± | 0.01 | 45.1 | ± | 2.5 | - | | |
| HS | 45 | 0-20 | 1 | 0.70 | - | 0.03 | 48.6 | ± | 3.5 | 15.98 | ± | 2.91 |
| HS | 68 | 0-20 | 2 | 0.68 | ± | 0.05 | 47.5 | ± | 4.5 | - | | |
| HS | 90 | 0-20 | 2 | 0.72 | ± | 0.01 | 49.9 | ± | 2.8 | - | | |
| KR | 0 | 0-20 | 1 | 0.98 | ± | 0.03 | 100.0 | ± | 2.9 | 32.66 | ± | 3.62 |
| KR | 2.5 | 0-20 | 1 | 1.02 | ± | 0.03 | 103.8 | ± | 4.2 | - | | |
| KR | 20 | 0-20 | 2 | 0.65 | ± | 0.02 | 66.6 | ± | 2.5 | - | | |
| KR | 30 | 0-20 | 1 | 0.75 | ± | 0.03 | 77.1 | ± | 3.6 | - | | |
| KR | 40 | 0-20 | 1 | 0.48 | ± | 0.01 | 49.1 | ± | 1.9 | 12.08 | ± | 2.43 |
| KR | 57 | 0-20 | 2 | 0.68 | ± | 0.04 | 69.1 | ± | 4.9 | - | | |
| KR | 98 | 0-20 | 2 | 0.43 | ± | 0.01 | 44.1 | ± | 1.7 | - | | |
| TW | 0 | 0-20 | 3 | 1.95 | ± | 0.06 | 100.0 | ± | 3.0 | 59.57 | ± | 6.12 |
| TW | 8.0 | 0-20 | 3 | 1.23 | ± | 0.04 | 63.3 | ± | 2.7 | 39.30 | ± | 4.61 |
| TW | 12 | 0-20 | 3 | 1.08 | ± | 0.02 | 55.3 | ± | 1.9 | 30.69 | ± | 2.79 |
| TW | 22 | 0-20 | 1 | 1.01 | ± | 0.03 | 51.8 | ± | 2.2 | - | | |
| TW | 32 | 0-20 | 2 | 1.07 | ± | 0.02 | 55.0 | ± | 2.0 | 30.46 | ± | 3.54 |
| TW | 40 | 0-20 | 3 | 0.81 | ± | 0.01 | 41.6 | ± | 1.4 | 24.07 | ± | 2.68 |
| TW | 60 | 0-20 | 3 | 0.70 | ± | 0.01 | 35.9 | ± | 1.3 | 15.91 | ± | 2.22 |
| TW | 90 | 0-20 | 2 | 0.58 | ± | 0.02 | 29.7 | ± | 1.4 | 15.09 | ± | 2.80 |
| HS | 0 | 20-40 | 1 | 0.08 | ± | 0.02 | 5.2 | ± | 1.6 | - | | |
| HS | 45 | 20-40 | - | 1.01 | ± | 0.03 | 145.0 | ± | 8.1 | - | | |
| TW | 0 | 20-40 | 2 | 0.11 | ± | 0.02 | 5.4 | ± | 1.0 | - | | |
| TW | 32 | 20-40 | 2 | 0.08 | ± | 0.01 | 7.8 | ± | 0.7 | - | | |
| TW | 40 | 20-40 | - | 0.29 | ± | 0.04 | 36.3 | ± | 4.4 | - | | |
| TW | 60 | 20-40 | - | 0.16 | ± | 0.04 | 23.0 | ± | 5.5 | *below detection limit* | | |

 **Sample labelling as presented in the main text includes the abbreviation of the sample agroecosystem, years of cultivation (YOC) and sampling depth interval: HS - Harrismith, KR - Kroonstad, TW - Tweespruit. The number *n* of**

**$^{239,240}$Pu replicate measurements includes both CologneAMS and ANU AMS measurement replicates. The quoted**
**plutonium activities from replicate measurements are weighted means and the uncertainties either dominated by AMS**
**counting statistics (weighted mean error) or external sources of uncertainty (standard error). For single measurements,**
**the 1σ measurement uncertainty provided by the AMS facilities dominates the final uncertainty. Percentual activities**
**of the topsoil samples are relative to the undisturbed reference sample for each agroecosystem. For the depth samples,**
**the percentage values denote the difference against the corresponding topsoil samples. $^{137}$Cs data uncertainties equal 1σ**
**measurement errors arising from μ spectrometry conducted at CSIRO. All $^{137}$Cs has been corrected for decay to**
**February 2012.**

**3.   Reply to Anonymous Referee #2**

Dear Anonymous Referee #2, thank you for your detailed review. In the following, we address the points you raised.

**AR2 #1: One minor point is that the introduction is to much on climate change, CO2 release etc. which is not the topic of this paper.**

Reply to AR2 #1: We have deleted ll. 50-52 and ll. 55-56.

**AR2 #2: On the other hand, nearly nothing is said on FRN use to assess wind erosion, methods to accomplish this or how to quantify erosion. The loss in SOC and FRN is attributed to wind erosion, just from visual assessment of the site. Could you please give exact slopes in the table for all sites? I find it hard to believe that at these altitudes you have so many sites completely flat. If this is true, this needs at least be identified. If it is not true, please consider that even small slopes will induce significant water erosion even on grasslands whenever you have rain events.**

Reply to AR2 #2: We refer to our replies to AR2 #10 and AR2 #11.

**AR2 #3: In some parts it is not clear to me, what the authors did with the data. They talk of „normalisation" but report data in percentages. Or was there really any normalisation done? Why not presenting the SOC concentrations and the FRN inventories over time? Much more interesting to soil scientists.**

Reply to AR2 #3: See reply to AR2 #21; we did not perform any normalisation but present just relative values, i.e. relative against the pooled grassland sample. Since the (maximum) absolute concentrations and inventories do vary in between the different agroecosystems, we present the data in this way. We also do so to facilitate the reader the comparison of our data with the data from the numerous previous studies published on our samples, which were presented in a similar fashion.

ll. 264: Rephrased for clarification: *We calculate the measured $^{239+240}$Pu inventories for all three agroecosystems to reflect relative inventory concentrations against the relevant pooled samples from the uncultivated plots. As a function of the duration of cultivation, a trend of initially decreasing activity with increasing cropping time is evident, although the rate of decline slows as time goes on (Fig. 2).*

**AR2 #4: To apply the FRN approach to assess soil erosion you need at least 3 if not more reference sites. You cannot assume, just because SOC and FRN is highest in your "natural" site, this would be a site without soil erosion. What you could do, however, is compare your arable sites to the one natural grassland and discuss if you have higher or lower erosion. Can you really guarantee that the natural sites were never ploughed since 1950s?**

Reply to AR2 #4: Our (sub-)samples taken from the reference sites are obtained from uncultivated grassland sites adjacent to the cultivated sites. For each agroecosystem, the reference sites were pooled to obtain a single composite sample. Hence, our three reference samples provide an average of all reference sites within an agroecosystem. As long as you accept interviews with the farmers and visual assessment of the samples taken with a steel cylinder as proof, we can guarantee that the natural sites were never ploughed.

**AR2 #5: 26 the sampled plots did not show signs of fluvial erosion? How do you assess this? And even if they do not show this today, how do you know for 100 years back?**

Reply to AR2 #5: We focused on the flat upland sites; the assessment was performed by visual interpretation and interviews with the farmers. From these interviews we also gathered that the older plots would even be less prone to erosion, because they usually are located closer to where the farmers settled, i.e. build their farms. Since their assessment of the landscape was crucial for their survival, we consider it as of even greater significance than our own assessment when we sample the sites.

**AR2 #6: 31 how do you know that 6% of the FRN inventory is lost in the first year?**

Reply to AR2 #6: From the exponential fit (eqn. 1).

**AR2 #7: 37 subtitles is misleading.... this is not about the release of CO2 (which is not the focus of your manuscript) but about the role of SOM in soils and how it is connected to erosion processes**

Reply to AR2 #7: We agree and have changed the title to:

l. 37: Soil organic matter and its degradation

**AR2 #8: 79-83 the discussion on the potential CO2 of African soils is not very convincing. You already have strong arguments why SOC loss is important: because of general soil degradation. I would suggest to leave that out.**

Reply to AR2 #8: We have deleted the text (ll. 79-83).

**AR2 #9: 93-96 I cannot follow your rational, why the molecular compound analysis will indicate SOM loss with increasing periods of cultivation**

Reply to AR2 #9: We agree and rephrase to:

l. 95: *A key finding of the study published by Lobe et al. (2001) was that SOM contents decreased exponentially with increasing periods of cultivation.*

**AR2 #10: Section 1.3. literature of how caesium or plutonium is used to estimate wind erosion is lacking. Web of science lists over 80 studies for caesium and 14 studies for plutonium. Also, no literature is discussed, how inventories are converted to soil erosion rates. As you obviously had a transition from natural grasslands (e.g., distinct depth profiles with FRN declining with depth) to ploughed arable soils (mixed plough layer) this is not a trivial task.**

Reply to AR2 #10: We have re-written introduction section 1.3. See also our reply to AR2 #13.

[revised manuscript text omitted]

**AR2 #11: Table 1: could you please give exact slopes in the table for all sites? I find it hard to believe that at these altitudes you have so many sites completely flat. If this is true, this needs at least be identified.**

Reply to AR2 #11: It is difficult to estimate the slope of the terrain, because the GPS location of the sampling sites is not as accurate as it could be achieved today. We kindly point out that the samples were taken in the late 1990s, and GPS selective availability was turned off in 2000; i.e. we face accuracy uncertainties on the order of 100 m at least. It is true that not the whole landscape is entirely flat; for example, the Highveld plateau is dissected by canyons. However, samples were taken from upland sites only (information added to l. 152 and l. 377). As an exercise, we calculated the surface slope for a 100 m buffer (100 m radius) around the GPS locations (i.e., ~30000 m²), as presented below. The data is based on a 30 m DEM, the highest resolution freely available for our sites. We note that the minimum slopes would relate to the upland sites:

| ID | Mean (°) | Min (°) | Max (°) |
|---|---|---|---|
| HS3.5 | 2.0 | 1.0 | 3.8 |
| HS8 | 1.7 | 1.4 | 1.9 |
| HS10 | 2.4 | 1.9 | 2.9 |
| HS30 | 2.1 | 1.8 | 2.2 |
| HS45 | 3.9 | 3.6 | 4.6 |
| HS68 | 3.0 | 1.1 | 3.9 |
| HS90 | 3.4 | 2.2 | 3.9 |
| KR2.5 | 0.5 | 0.0 | 1.1 |
| KR20 | 0.9 | 0.4 | 1.1 |
| KR30 | 0.7 | 0.0 | 1.1 |
| KR40 | 0.8 | 0.2 | 1.1 |
| KR57 | 0.7 | 0.0 | 1.3 |
| KR98 | 0.6 | 0.0 | 1.1 |
| TW8.5 | 1.5 | 1.0 | 1.9 |
| TW12 | 1.1 | 0.7 | 1.5 |
| TW22 | 1.1 | 0.3 | 1.7 |
| TW32 | 1.7 | 0.8 | 2.1 |

| | | | |
|---|---|---|---|
| TW40 | 1.8 | 1.1 | 1.9 |
| TW60 | 1.6 | 1.0 | 2.1 |
| TW90 | 1.6 | 1.4 | 2.1 |

**AR2 #12: 127-129 this is a crude oversimplification. As FRN is deposited with wet and dry deposition, you have substantial heterogeneity. This needs to be considered in taking a adequate number of reference cores. These reference cores should have a CV < 30%. See Sutherland et al.....**

Reply to AR2 #12: We agree. Hence, we have focused on individual agroecosystems, and the reference values are obtained from pooled samples providing an average characterisation of each agroecosystem (n = 5 subsamples per individual site within each agroecosystem). The heterogeneity is a further argument why we show inventories relative to the pooled reference sites in Figure 2. We deleted the sentence in ll. 128-129 and added instead (please also see reply to AR2 #10):

l. 128: *However, variability in wet and dry fallout deposition as well as microtopography even on the local scale needs to considered. Thus, a rather short distance between the reference site and the eroding site and a large number of subsamples to characterise the reference site FRN inventory (i.e. n >10; Sutherland, 1996) are considered as crucial (Sutherland, 1996; He and Walling, 1996; Van Pelt, 2013).*

l. 155: Rephrased and added for clarification: *Each of the three datasets includes one composite sample (HS0, KR0, TW0) taken from native grassland sites located directly adjacent to the respective cultivated sites. These reference samples represent the amalgamated sample material from all grassland sites within a common agroecosystem.*

ll. 428-429: Changed to: *For FRNs, such differences are likely to arise from the spatially variable deposition patterns, given the distances between the different agroecosystems of about 100-300 km. However, grain size data also indicate an increase in the silt fraction towards the south-east (Fig. S1).*

**AR2 #13: Section 2.1 belongs to introduction. Together with a discussion on the use of FRN to estimate wind erosion.**

Reply to AR2 #13: We have shifted the section to the introduction (section 1.3); the combined piece of text is presented in our reply to AR2 #10.

**AR2 #14: 153 how flat is flat? Already very slight slopes will induce water erosion in African soils. If you have heavy rain events after dry periods, slopes of <2° might already induce water erosion.**

Reply to AR2 #14: See our reply to AR1 #11.

**AR2 #15: 174 why <20 μm? In Africa, you can expect to have winds which blow out larger grains.....?**

Reply to AR2 #15: In fact, we did measure the bulk soil (see above). We apologise for the confusion caused.

**AR2 #16: Did you not do any decay correction for the caesium to a reference year? Why not? How do you then relate to the year of deposition?**

Reply to AR2 #16: All caesium data is decay-corrected to February 2012 (see Fig. 4 caption). To clarify, we add:

l. 198: All $^{137}$Cs data presented in this publication have been decay-corrected to February 2012 (the time of measurement).

**AR2 #17: Table 2: this table looks like a working table from the lab. Could you please make it reader friendly with a column for site name, sampling depth etc???**

Reply to AR2 #17: We have updated the table accordingly (see above).

**AR2 #18: 240 – 255 most of this is redundant if you format Table 2 properly, make suitable headings and explain some of this in the methods. This has nothing to do with results.**

Reply to AR2 #18: We agree. We deleted ll. 240-245, ll. 247-249, and incorporated ll. 245-246 in l. 234. Furthermore, we shifted ll. 249-255 to l.259.

l. 234: Content from ll. 245-246 added: *The latter are reflected by the standard error, i.e. the standard deviation of the set of measurements divided by the square root of the number of measurements σ/√n (the larger uncertainty value was chosen for each sample).*

**AR2 #19: 260 -263 sentences incomprehensible**

Reply to AR2 #19: We assume your comment arises from our insufficient description of the reference site samples. See our reply to AR2 #12.

**AR2 #20: 264 sentences like "Figure 2A shows…" are unnecessary… please give adequate figure headings and delete these kind of sentences**

Reply to AR2 #20: We have rephrased the sentence, see our reply to AR2 #3.

**AR2 #21: Figure 2: what do you mean by "normalised"? What you did is, setting the site you defined as natural to 100% and calculated percentages from that. But this is not normalised? Why do you give percentages and not the original concentrations? Or did you any other normalisation with the data?**

Reply to AR2 #21: Absolutely correct. We did not do any other normalisation with the data. Hence, we changed the title of the ordinate. See updated Fig. 2 above and our reply to AR2 #3.

**AR2 #22: 275 what do you mean by "internally consistent"?**

Reply to AR2 #22: The data of both ANU and CologneAMS is consistent. We deleted "internally" (l. 275).

**AR2 #23: 290 even higher**

Reply to AR2 #23: Corrected; thank you.

**AR2 #24: Section 3.3 It is not unusual that Pu migrates down to 20-30 cm or even 35. But this might also indicate deposition of soil material. As such, you need reference soil profiles to compare you original FRN deposition to erosional or depositional sites.**

Reply to AR2 #24: We agree (AR1 raised similar concerns). Here is how we adjusted our text:

l. 170: *A clear disadvantage of the applied sampling scheme is the lack of high-resolution depth profile samples, which was not required for the originally intended sample analyses. As a consequence, we are unable to present FRN mass depth profile data, and thus cannot reasonably infer mass redistribution rates as typically presented in FRN studies (e.g. Alewell et al. 2014, Lal et al. 2013, Meusburger et al. 2018).*

l. 236: peak plutonium deposition added and rephrased: *Since our sampling strategy included a spatial averaging of sampling material from each plot investigated (Sect. 2.2), the best explanation for the elevated plutonium activity in HS45/20-40 may be related to a former ploughing to 40 cm near in time to peak plutonium deposition that was not recorded during farmers' interviews or to sample contamination. Elevated inventories measured in two further depth samples might point to a certain degree of leakage of plutonium towards greater depths, but not necessarily in the pre-fallout soils (Sect. 4.3). We also note that in a reasonably comparable setting (Bsh climate; Big Spring, USA), $^{137}Cs$ and $^{239+240}Pu$ concentrations dropped sharply below the $A_p$ horizon in an Aridic Paleustalf that had been cultivated since 1915 (Van Pelt et al., 2007; Van Pelt and Ketterer, 2013).*

l. 314: *However, any leakage could equally affect the isotope concentrations and may thus not be reflected by the ratio.*

l. 371: leakage added: *Likewise, a possible incorporation of Pu-marked plant material into the soil column after harvesting might have contributed to elevated inventories found in the three depth samples with cropping histories exceeding 35 years (HS45/20-40, TW40/20-40, and TW60/20-40). In case of wet fallout deposition, such an enhanced downward migration could also have been promoted by the physical disturbance of the ploughed soil (cf. Das Gupta et al., 2006). These factors appear not to have been significant for the plot that was converted after the peak episode of global fallout (TW32/20-40). However, the topsoil samples TW40/0-20 and TW60/0-20 had the strongest inventory losses of all samples (Fig. 2a; Table 2), and show the highest negative residues against the exponential model. Thus, and given the general low scatter of the post-35 YOC data points, we may argue that if significant migration of Pu-marked soil particles below 0-20 cm has occurred, the two samples in question could represent the cases of maximum leakage in our dataset.*

l. 418: leakage added: *Furthermore, a certain degree of leakage of plutonium-marked particles to greater depths could contribute to a lowering of $^{239+240}Pu$ activities in the topsoil over time.*

l. 423: rephrased to: *To resolve this issue, longer timescales need to be monitored and high-resolution depth profiles sampling implemented in future studies.*

**AR2 #25: 323 to apply the FRN approach to assess soil erosion you need at least 3 if not more reference sites. You can not assume, just because SOC and FRN is highest, this would be a site without soil erosion. What you could do, however, is compare your arable sites to the one natural grassland and discuss if you have higher or lower erosion.**

Reply to AR2 #25: We kindly refer to our replies AR2 #4 and AR2 #12.

**AR2 #26: 336 I can not follow this assumption nor the conclusion**

Reply to AR2 #26: We agree, there is no reasonable relationship between the two clauses. We have deleted the first part of the sentence (ll. 336-337).

**AR2 #27: 341 do not understand why exponential decline indicates higher adsorption and the conclusion to aggregation seems far fetched (or explain and constrain it better)**

Reply to AR2 #27: The exponential model predicts the retention of Pu over the long term. However, we state that it may not be the most accurate model to chose (ll. 420-423). Note that the original sentence was modified according to the reviewer's comments (see below). As for the aggregation, we find it reasonable to present our ideas.

ll. 420-423: *Instead, a bi-exponential model as suggested to predict SOM decrease over time (e.g. Amelung et al., 2002; Lobe et al., 2001; Lobe et al., 2011) could reflect the long-term fate of $^{239+240}Pu$ inventories in the topsoils more accurately. To resolve this issue, longer timescales need to be monitored and high-resolution depth profiles sampling implemented in future studies.*

**AR2 #28: Section 4.3 I do not agree with this discussion. Of course, using FRN in Southern Hemisphere means you can only assess the period from 1950ties to now. However, soils which were already ploughed during that time, are the only ones where you could quantify soil erosion, as you do not need any reference site depth profile but only a total inventory of the reference site and can then apply the proportional model. For all other sites, e.g. the sites changing from natural to ploughed in between, you would have to assume an (unknown) depth profile first and then a mixed plough layer after to quantify erosion.**

Reply to AR2 #28: We have measured composite depth samples (20-40 cm), that clearly to indicate that there was – if any – very limited migration of Pu below 20 cm in the pooled reference sites (Fig. 5). We take that as a sufficient indication for assuming that the Plutonium was predominantly stored in the 0-20 cm soil column.

**AR2 #29: 402 What do you mean "deflation processes rather than turnover rates"?**

Reply to AR2 #29: Lobe et al. (2001) identified the turnover (predominantly mineralisation) of SOM in the silt and clay fractions as the main factor controlling the time-dependent decrease of SOM stocks in the investigated arable soils. Here we postulate that deflation processes may play a more important role for the decrease in SOM contents (see Fig. 2C).

**AR2 #30: Figure 5: I am generally puzzled by this Figure. If you see this strong decline in Pu concentrations over time and attribute this to erosion process, this means that what you measure today as 0-20 cm depth was 20-40 cm depth 100 years ago, right? But why don't you see any changes in 20-40 cm depth? This should for sure decline to zero 60 years after cultivation. I think there is something else going on and you should for sure calculate your inventories considering your mass depth of soil and may be even assess erosion rates comparing it to you natural site. These simulated erosion rates (which, strictly speaking would not be absolute erosion rates but rates above the natural site values) would then give you some confidence about possible processes going on. However, you clearly need depth profiles of FRN from your natural site and the time of conversion from natural to arable land.**

Reply to AR2 #30: As noted above, we have calculated inventories now; however, we cannot supply depth profiles. A certain degree of vertical migration may be affecting the FRN inventories in the 0-20 cm soil column (see our replies to AR2 #24). Concerning the erosion of soil, we kindly point out that a significant fraction of Pu loss may be due to the deflation of SOM, and not the clastics (see Fig. 2C).

**AR2 #31: Conclusion – Sorry to say, but from the above, I can not really see that these conclusions are justified by your data.**

Reply to AR2 #31: No problem, that is what a good reviewing process is for. We thank you again for your comments, apologise for the wrong grain size stated and interpreted in the initially submitted manuscript, and hope that you are happier with our rewritten conclusion. Furthermore, we invite you to read our reply to AR1, which has also brought significant change to the manuscript. For the new version of our conclusion, we have shifted bits of text from section 4.5 to the conclusion and deleted section 4.5 afterwards. The new version of the conclusion:

[revised manuscript text omitted]

---

## Author Response (AR2)

Mohren et al. Biogeosciences manuscript – Reply to Anonymous Referee #3

Dear Anonymous Referee #3, thank you for reviewing our manuscript and for providing your detailed
assessment as well as the additional references. We address the points you raised below.

**AR3 #1: In the manuscript, the authors recognise (Line 465) the occurrence of deposition from**
**wind erosion and dust emission elsewhere. The source of that deposition could be from nearby**
**and therefore include a wide range of particle sizes (coarse material will not be preferentially**
**removed with distance). Alternatively, that deposited material may be distal and therefore**
**associated with fine material very likely enriched in fallout radionuclides. In their approach to**
**establishing a reference site, the authors have neglected to consider that their chosen site may be**
**influenced by deposited aeolian material. A larger reference inventory would change the**
**magnitude of losses and gains identified at sites made relative to that site. The results are therefore**
**uncertain depending on the amount of deposited aeolian material at the native grassland site by**
**contrast to deposition at the other sites.**

**The corollary is that deposited aeolian material is likely to be occurring across the region as the**
**authors suggest (Line 465). If different sources of that deposited material are proximal and distal,**
**then the deposited particle size distribution will change and consequently the FRN concentration**
**per unit mass will change. These changes in concentrations do not conform to the expected**
**behaviour of the approach and its underlying assumptions (Chappell, 1999). The uncertainty is**
**further dependent on the mixing of deposited material from different sources.**

Reply to AR3 #1: We agree and add and/or change text as follows:

l. 168: *Furthermore, sampling focused on upland agricultural areas with level surfaces to minimise the*
*possibility of fluvial erosion and aeolian influx affecting the SOM content (and FRN concentrations).*

l. 193: *Likewise, we cannot assess effects of soil particle fluxes that may alter inventories in the*
*composite reference samples (cf. Chappell, 1999; Sect. 4.4).* (see also AR3 #2)

l. 325: *Exceeding $^{239+240}$Pu has been proposed to reflect grain-size dependent preferential adsorption*
*patterns (e.g. Everett et al., 2008, Xu et al., 2017), and such a pattern could become important in case*
*of selective erosion or soil particle influx.*

l. 474: *If such regional patterns of sediment redistribution caused significant influx of soil particles to*
*reference sites from both local and regional sources after global fallout (cf. Wiggs and Holmes, 2010),*
*it is possible that FRN inventories have been subject to alterations. As our methodological approach*
*relies on undisturbed reference sites, significant influx to the reference sites would violate that most*
*important precondition. Chappell (1999) showed that influx of soil particles can significantly alter $^{137}$Cs*
*specific activities at a reference plateau site located in semi-arid bushland. Influx of distal dust particles*
*may increase FRN concentrations (Chappell, 1999), more proximal influx of coarser grains could dilute*
*them (Funk et al., 2011). Hence, post-fallout accumulation of soil particles on our reference sites could*
*have different effects on the overall FRN concentrations, depending on the concentration of FRNs in the*
*deposited soil particles (generally linked to soil particle grain size and source). However, visual*
*inspection of the reference sites before sampling suggested that significant coarse-grained influx from*
*local sources can be ruled out. In addition, Funk et al. (2011) demonstrated that $^{137}$Cs reference sites*
*rather unaffected by aeolian deposition could be identified in their study region, which resembles our*
*study setting (grassland plateau site in Mongolia with significant wind erosion). We also note that soil*
*bulk densities are generally homogenous across the individual agroecosystems and that the $^{137}$Cs and*
*$^{239+240}$Pu concentrations we obtained are strongly correlated (Fig. 3). Given that $^{137}$Cs and $^{239+240}$Pu are*
*suspected to show somewhat different grain size and SOM-dependent adsorption patterns (Sect. 1.3),*

*the finding could imply that influx to the reference sites was limited. Furthermore, given that reference*
*sites were located directly adjacent to the eroding sites, alterations of the relative inventories due to soil*
*particle influx should decrease in significance at decreasing YOCs.*

**AR3 #2: The authors have provided a very basic description of the assumptions upon which the**
**fallout radionuclides are used to estimate soil redistribution (around Line 125). I think a much**
**clearer description of the assumptions is required. I think this description needs to be updated**
**with the alternative approach using resampling (Li et al., 2011). Most importantly, I think it is**
**essential that this description in the manuscript is improved by including a more critical**
**evaluation of the approach including the work by Foster and Parsons (2011), the Comment by**
**Mabit et al. (2011) and other arising commentary since then.**

Reply to AR3 #2: We add and/or change text as follows:

l. 124: *The method to assess soil redistribution by using FRN concentrations relies on several*
*assumptions which should be met (for an overview, see e.g. Van Pelt, 2013; Zapata, 2002; a critical*
*assessment of the technique and a reply to the critical view are provided by Parsons and Foster, 2011,*
*and Mabit et al., 2013, respectively). One precondition of the widely used traditional sampling approach*
*(cf. Li et al., 2011) is that of a homogeneous distribution of the target FRN over the limited area covering*
*the undisturbed reference site and the nearby eroding sites.*

l. 130: *A certain variance attached to reference inventories may be inevitable but can be reduced by*
*applying the repeated-sampling-approach, which relies on on-site point-specific reference inventories*
*(Li et al., 2011; Kachanoski and De Jong, 1984). Such a sampling strategy, however, would require a*
*resampling campaign and hence be difficult to implement in our case given possible changes in land*
*use and cropping practices since 1998 as well as individual permits required.*

*A reasonable application of the traditional approach relies on reference sites that ideally are vegetated*
*with perennial grass or low herb cover (Pennock and Appleby, 2002) and shielded from sediment*
*deposition, such as likely achieved on level upland sites (Funk et al., 2011).*

l. 135: *While $^{137}Cs$ sorption has been found to be generally dependent on the availability of cation*
*exchange sites in soils and hence on clay mineralogy (Mabit et al., 2013; Parsons and Foster, 2011), it*
*might bind more selectively to the clay fraction compared to plutonium, implying that $^{137}Cs$ could be*
*more sensitive to preferential transport (Xu et al., 2017).*

l. 182: *Up to nine different agricultural plots were sampled per agroecosystem, with the requirement*
*that the cultivation history (up to 98 yrs) could be precisely ascertained and a reference site could be*
*sampled adjacent to the eroding site (cf. Lobe, 2003).*

l. 185: *The latter were included in order to test whether the topsoil sampling approach captured most*
*of the plutonium stored within the soil column (cf. Parsons and Foster, 2011).*

L. 190: *The sampling scheme, which originally did not focus on FRN analyses, has some disadvantages*
*potentially biasing FRN data interpretation (cf. Sect. 1.3). Firstly, the lack of high-resolution depth*
*profile samples means that we are unable to present FRN mass depth profile data. Consequently, we*
*cannot reasonably infer mass redistribution rates as typically presented in FRN studies (e.g. Alewell et*
*al., 2014; Lal et al., 2013; Meusburger et al., 2018). Likewise, we cannot assess effects of soil particle*
*fluxes that may alter inventories in the composite reference samples (cf. Chappell, 1999; Sect. 4.4),*
*although visual inspection suggests that significant coarse-grained influx from local sources can be*
*ruled out. Finally, amalgamation of the reference site samples (n = 7-9 samples per agroecosystem;*
*with n = 5 subsamples per site) implies that we cannot provide statistical measures to evaluate the*
*accuracy of fallout inventories in the reference samples.*

l. 303: *In order to investigate whether plutonium could have migrated below this soil layer (e.g. Parsons and Foster, 2011), samples spanning the depth interval 20-40 cm were analysed for selected sites from the Tweespruit (n = 4) and Harrismith (n = 2) agroecosystems.*

**AR3 #3: The abstract does not adequately represent the issue that the identification of wind erosion needs to be without the presence of water erosion. I think this topic is reasonably well described in the main text, perhaps with the inclusion of Van Pelt et al. (2017).**

Reply to AR3 #3: We rephrase as sentence in the abstract and add the reference (l. 149 & 152).

l. 26-28: *Wind erosion has previously been shown to play a dominant role in soil particle loss from agricultural sites in the Highveld, and the level plots we investigate here did not show any evidence of fluvial erosion. Hence, we interpret the fate of soil fines, including SOM, to be governed by wind erosion.*

**AR3 #4: The word 'flat' is used throughout the manuscript to incorrectly describe how level the land surface is. The word flat is a description of the land surface roughness and should be replaced with the word level, as appropriate.**

Reply to AR3 #4: We follow your advice and change the terms accordingly (l. 81, 97, 130, 169).

**AR3 #5: Line 165 "Since our samples were already taken in 1998 and have been characterised in numerous studies (Amelung et al., 2002; Lobe..." This sentence is ambiguous in the description of the characterisation. Please rephrase.**

Reply to AR3 #5: We rephrase:

l. 165: *The samples analysed in this study were taken in 1998 and splits from these samples have been measured in previous studies to investigate a variety of soil components and patterns of soil degradation over time (Sect. 1.2). In the following, we give a brief overview of the sampling strategy that was applied.*

**Other changes made to the manuscript**

Words or letters added/removed:

l. 18: losses → *loss*

l. 22: + *during the*

l. 87: cropping → *cultivation (YOC)*

l. 137: regime → *regimes*

l. 305: + *indicate*

l. 360, 365, 368, 404, 421, 424: years of cropping → *YOC*

l. 408: + *1.2 and*

l. 469: are likely to arise from the → *could arise from*

l. 517: + *We thank three anonymous referees, whose comments have significantly improved the quality of this paper.*

Section titles changed:

l. 164: 2.1 Sampling strategy and sample processing → *2.1 Sampling strategy*

l. 194: + *2.2 Sample processing*

l. 223: 2.2 FRN measurements → *2.3 FRN measurements*

l. 246: 2.3 Interpretation of $^{239+240}$Pu results → *2.4 Interpretation of $^{239+240}$Pu results*

l. 374: 4.3 Temporal limitation of $^{239+240}$Pu topsoil inventories → *4.3 Factors that may influence the*
*interpretation of $^{239+240}$Pu topsoil inventories*

**References**

Alewell, C., Meusburger, K., Juretzko, G., Mabit, L., and Ketterer, M. E.: Suitability of $^{239+240}$Pu and
$^{137}$Cs as tracers for soil erosion assessment in mountain grasslands, Chemosphere, 103, 274-280,
https://doi.org/10.1016/j.chemosphere.2013.12.016, 2014.
Chappell, A.: The limitations of using 137Cs for estimating soil redistribution in semi-arid
environments, Geomorphology, 29, 135-152, https://doi.org/10.1016/S0169-555X(99)00011-2, 1999.
Everett, S. E., Tims, S. G., Hancock, G. J., Bartley, R., and Fifield, L. K.: Comparison of Pu and 137Cs as
tracers of soil and sediment transport in a terrestrial environment, Journal of Environmental
Radioactivity, 99, 383-393, https://doi.org/10.1016/j.jenvrad.2007.10.019, 2008.
Funk, R., Li, Y., Hoffmann, C., Reiche, M., Zhang, Z., Li, J., and Sommer, M.: Using 137Cs to estimate
wind erosion and dust deposition on grassland in Inner Mongolia-selection of a reference site and
description of the temporal variability, Plant and Soil, 351, 293-307, 10.1007/s11104-011-0964-y,
2011.
Kachanoski, R. G. and de Jong, E.: Predicting the Temporal Relationship between Soil Cesium-137 and
Erosion Rate, Journal of Environmental Quality, 13, 301-304,
https://doi.org/10.2134/jeq1984.00472425001300020025x, 1984.
Lal, R., Tims, S. G., Fifield, L. K., Wasson, R. J., and Howe, D.: Applicability of 239Pu as a tracer for soil
erosion in the wet-dry tropics of northern Australia, Nuclear Instruments and Methods in Physics
Research Section B: Beam Interactions with Materials and Atoms, 294, 577-583,
https://doi.org/10.1016/j.nimb.2012.07.041, 2013.
Li, S., Lobb, D. A., Kachanoski, R. G., and McConkey, B. G.: Comparing the use of the traditional and
repeated-sampling-approach of the 137Cs technique in soil erosion estimation, Geoderma, 160, 324-
335, 10.1016/j.geoderma.2010.09.029, 2011.
Lobe, I.: Fate of organic matter in sandy soils of the South African Highveld as influenced by the
duration of arable cropping, Bayreuther bodenkundliche Berichte, 79, Lehrstuhl für Bodenkunde und
Bodengeographie der Univ. Bayreuth, Bayreuth2003.
Mabit, L., Meusburger, K., Fulajtar, E., and Alewell, C.: The usefulness of 137Cs as a tracer for soil
erosion assessment: A critical reply to Parsons and Foster (2011), Earth-Science Reviews, 127, 300-
307, 10.1016/j.earscirev.2013.05.008, 2013.
Meusburger, K., Porto, P., Mabit, L., La Spada, C., Arata, L., and Alewell, C.: Excess Lead-210 and
Plutonium-239+240: Two suitable radiogenic soil erosion tracers for mountain grassland sites,
Environ Res, 160, 195-202, 10.1016/j.envres.2017.09.020, 2018.
Parsons, A. J. and Foster, I. D. L.: What can we learn about soil erosion from the use of 137Cs?, Earth-
Science Reviews, 108, 101-113, 10.1016/j.earscirev.2011.06.004, 2011.
Pennock, D. and Appleby, P.: Site selection and sampling design, in: Handbook for the assessment of
soil erosion and sedimentation using environmental radionuclides, Springer, 15-40, 2002.
Van Pelt, R. S.: Use of anthropogenic radioisotopes to estimate rates of soil redistribution by wind I:
Historic use of 137Cs, Aeolian Research, 9, 89-102, 10.1016/j.aeolia.2012.11.004, 2013.

Wiggs, G. and Holmes, P.: Dynamic controls on wind erosion and dust generation on west-central
Free State agricultural land, South Africa, Earth Surface Processes and Landforms, 36, 827-838,
https://doi.org/10.1002/esp.2110, 2010.
Xu, Y., Pan, S., Wu, M., Zhang, K., and Hao, Y.: Association of Plutonium isotopes with natural soil
particles of different size and comparison with 137Cs, Science of The Total Environment, 581-582,
541-549, https://doi.org/10.1016/j.scitotenv.2016.12.162, 2017.
Zapata, F.: Handbook for the assessment of soil erosion and sedimentation using environmental
radionuclides, Springer2002.